# Learning to Price with Resource Constraints: From Full Information to Machine-Learned Prices

**Ruicheng Ao**
MIT
aorc@mit.edu

**Jiashuo Jiang**
HKUST
jiang@ust.hk

**David Simchi-Levi**
MIT
dslevi@mit.edu

## Abstract

Dynamic pricing with resource constraints is a critical challenge in online learning, requiring a delicate balance between exploring unknown demand patterns and exploiting known information to maximize revenue. We propose three tailored algorithms to address this problem across varying levels of prior knowledge: (1) a Boundary Attracted Re-solve Method for the full information setting, achieving $O(\log T)$ regret without imposing additional conditions; (2) an online learning algorithm for the no information setting, delivering an optimal $O(\sqrt{T})$ regret; and (3) an estimate-then-select re-solve algorithm for the informed price setting, leveraging machine-learned prices with known error bounds to bridge the gap between full and no information scenarios. Moreover, through numerical experiments, we demonstrate the robustness and practical applicability of our approaches. This work advances dynamic pricing by offering scalable solutions that adapt to diverse informational contexts while relaxing classical assumptions.

## 1 Introduction

Dynamic pricing is a classical problem in online learning and decision-making. With the growth of e-commerce, there has been an increasing focus on the development of efficient dynamic pricing policies in the recent literature, see for example Amin et al. [2014], Javanmard and Nazerzadeh [2019], Shah et al. [2019], Cohen et al. [2020], Xu and Wang [2021], Bu et al. [2022], Xu and Wang [2022], Xu et al. [2025].

Dynamic pricing is challenging due to the unknown relationship between the posted price and the corresponding demand. Usually, one can model the dynamic pricing problem as a bandit problem to balance the exploration of the price-demand relationship and the exploitation of setting the optimal price, where the price option is regarded as an arm and the corresponding revenue (demand multiplied price) is regarded as the reward for the arm. However, different from the classical multi-arm-bandit model where there is a finite number of arms, the price option can be infinite and continuous, which brings significant challenge to the learning part.

Another challenge to dynamic pricing is the existence of resource constraints. In practice, a product usually enjoys a fixed initial inventory to be sold during the horizon and the pricing decision is made to balance the demand such that the total revenue is maximized subject to the resource constraints. Note that the existence of resource constraints will significantly complicate the problem because the optimal decision is no longer to maximize the revenue but to balance the revenue versus resource consumption in an optimal way. Past work on dynamic pricing mainly focuses on dealing with the learning challenge, but ignores the challenge of resource allocation (e.g. Keskin and Zeevi [2014]). In our paper, we address this fundamental issue by developing near-optimal learning policies for dynamic pricing problems with resource constraints.

### 1.1 Preliminaries

To study this problem, we consider a setting with $m$ resources and $n$ products over a horizon of $T$ periods. At each period $t$, the decision-maker (DM) sets a price vector $\boldsymbol{p}^t \in [L, U]^n$, observes a

stochastic demand $\boldsymbol{d}^t \in \mathbb{R}_+^n$, and then updates the remaining capacities according to

$$\boldsymbol{c}^{t+1} = \boldsymbol{c}^t - A\,\boldsymbol{d}^t,$$

where $A \in \mathbb{R}_+^{m \times n}$ is a known consumption matrix and $\boldsymbol{c}^0$ is the initial resource capacity vector. The objective is to choose a pricing policy that maximizes the total expected revenue

$$\mathbb{E}\Big[ \sum_{t=1}^{T} (\boldsymbol{p}^t)^\top \boldsymbol{d}^t \Big],$$

while never violating the resource constraints (i.e., not serving demand that exceeds the remaining capacity).

Rather than compare to the intractable stochastic optimum of this problem, we evaluate any policy $\pi$ via its *regret* relative to an idealized *fluid benchmark*. The fluid benchmark replaces random demand by its expectation and computes the optimal solution of the resulting deterministic problem. In particular, assuming a known demand function $f(\boldsymbol{p}) = \mathbb{E}[d \mid p]$, the fluid optimal value is given by the solution of

$$V^{\text{Fluid}}(\boldsymbol{c}^0) = \max_{p \in [L,U]^n} \{ T \cdot \boldsymbol{p}^\top f(\boldsymbol{p}) \mid A\,f(\boldsymbol{p}) \le \boldsymbol{c}^0/T \}, \tag{1}$$

which is a deterministic program. To ensure the problem is well-posed, we make the following assumptions:

**Assumption 1.1** (Linear demand model)**.** We assume the true demand function is linear. That is, there exist parameters $\boldsymbol{\alpha} = (\boldsymbol{\alpha}_1, \dots, \boldsymbol{\alpha}_n)^\top \in \mathbb{R}^n$ and $B \in \mathbb{R}^{n \times n}$ such that

$$f(\boldsymbol{p}) = \boldsymbol{\alpha} + B\,p$$

for any price vector $p$ in the feasible domain.

Although we focus on a linear demand model for clarity, our analysis can be extended to more general parametric demand models under standard regularity conditions (e.g., Lipschitz continuity and concavity). We also impose a mild condition ensuring the demand slope is strictly negative-definite:

**Assumption 1.2** (Negative definiteness)**.** The matrix $B$ is negative definite; namely, $\lambda_{\max}(B+B^\top) < 0$ where $\lambda_{\max}(\cdot)$ denotes the largest eigenvalue of a matrix. Moreover, we assume that $\alpha$ is large enough to guarantee $f(\mathbf{p}) \ge 0$ for any price vector $\mathbf{p}$ in the feasible domain.

This condition implies that each price has a strictly decreasing effect on demand and that $B$ is invertible. (Indeed, if $B\boldsymbol{p} = 0$ for some nonzero $p$, then $(\boldsymbol{p})^\top (B + B^\top)p = 0$, contradicting $\lambda_{\max} < 0$.) In particular, the revenue function $p \mapsto (\boldsymbol{p})^\top f(\boldsymbol{p}) = (\boldsymbol{p})^\top \boldsymbol{\alpha} + (\boldsymbol{p})^\top B\boldsymbol{p}$ is concave in $p$ under this assumption.

To model demand uncertainty, we assume the observed demand includes independent noise. Formally, at each period $t$ the realized demand is

$$\boldsymbol{d}^t = f(\boldsymbol{p}^t) + \boldsymbol{\epsilon}^t,$$

where $\boldsymbol{\epsilon}^t \in \mathbb{R}^n$ is a random noise vector with $\mathbb{E}[\boldsymbol{\epsilon}^t] = 0$ and $\epsilon^t \ge -f(p^t)$. We assume $\boldsymbol{\epsilon}^t$ is sub-Gaussian with variance parameter $\sigma^2$, meaning $\mathbb{P}(|v^\top \boldsymbol{\epsilon}^t| \ge \lambda) \le 2\exp(-\lambda^2/(2\sigma^2))$ for any unit vector $v$. The revenue collected at time $t$ is $r^t = \boldsymbol{p}^{t\top} \boldsymbol{d}^t$, and the resource consumption is $A\,\boldsymbol{d}^t$, so the capacity updates as $\boldsymbol{c}^{t+1} = \boldsymbol{c}^t - A\,\boldsymbol{d}^t$. (If fulfilling the demand $\boldsymbol{d}^t$ for some product would exhaust a resource, we assume the sales of that product are curtailed to respect the capacity constraint.) We define a filtered probability space $(\Omega, \mathcal{F}, \{\mathcal{F}^t\}_{t=0}^{T}, \mathbb{P})$ where $\mathcal{F}^t$ represents all information available up to time $t$. We have an upper bound on the expected revenue of any admissible policy for the original stochastic problem .

**Proposition 1.3** ([Gallego and Van Ryzin, 1994])**.** *For any policy $\pi$,*

$$V^{\text{Fluid}}(\boldsymbol{c}^0) \ge \mathbb{E}\Big[ \sum_{t=1}^{T} (\boldsymbol{p}^t)^\top \boldsymbol{d}^t \Big].$$

Thus, $V^{\text{Fluid}}(\boldsymbol{c}^0)$ is a natural yardstick for performance. We define the regret of a policy $\pi$ over horizon $T$ as

$$\text{Regret}_T(\pi) = V^{\text{Fluid}}(\boldsymbol{c}^0) - \mathbb{E}\Big[ \sum_{t=1}^{T} (\boldsymbol{p}^t)^\top \boldsymbol{d}^t \Big].$$

## 1.2 Main Results and Contributions

In this paper, we design algorithms that attain near-optimal performance (low regret) in three different information settings:

1. **Full-information setting.** When the demand function $f(\boldsymbol{p})$ is fully known to the seller, we propose a *Boundary-Attracted Re-solve* method (Algorithm 1) that achieves $O(\log T)$ regret. Notably, our approach does *not* require the restrictive non-degeneracy condition (unique fluid optimal dual prices) assumed in prior work [Wang and Wang, 2022, Li and Ye, 2022], making it robust even when the fluid LP has multiple optima (a common occurrence in practice; see Bumpensanti and Wang 2020).

2. **No-information setting.** When the demand function is initially unknown, we develop an online learning algorithm (Algorithm 2) that interleaves exploration with re-solving. Our algorithm attains the optimal $O(\sqrt{T})$ regret rate, matching the known lower bound for this problem [Keskin and Zeevi, 2014]. This guarantees effective pricing decisions despite the initial uncertainty about demand.

3. **Informed-price setting.** In the intermediate case, the seller has access to an initial "machine-learned" price-demand data point with a known error bound. We design a novel *estimate-then-select* algorithm (Algorithm 3) that adaptively leverages this informed price when it is reliable and falls back to exploration when it is not. We prove that it achieves a regret of $O(\min\{\sqrt{T}, (\epsilon^0)^2 T\} + \log T)$, where $\epsilon^0$ is the known error bound of the initial estimate. In particular, with a sufficiently accurate initial price estimate, the regret improves beyond the $\Omega(\sqrt{T})$ rate of uninformed learning, effectively bridging the gap between the full-information and no-information extremes.

In summary, our work provides a unified framework for dynamic pricing under resource constraints that relaxes several unrealistic assumptions and incorporates offline information into online learning. By addressing all three information regimes, we remove prior limitations (such as requiring unique fluid solutions or having no access to historical data) and develop algorithms that are theoretically optimal and practically implementable. Finally, we validate the scalability and robustness of our methods through extensive numerical experiments (see Appendix 5 for details), demonstrating that they perform well in multi-resource settings and under various levels of noise and model misspecification.

## 1.3 Related Literature

Our work connects to four main research streams: (i) bandits with knapsacks; (ii) online resource allocation with fluid approximations; (iii) dynamic pricing with offline data and parameter estimation; and (iv) learning-augmented algorithms.

**Bandits with knapsacks.** The BwK framework, introduced by Agrawal and Devanur [2016], addresses online decision-making with resource constraints. Two main approaches have been developed. The first approach selects an optimal randomized policy from a finite policy class [Badanidiyuru et al., 2014]. Using this method, Agrawal et al. [2016] achieve $O(\sqrt{T})$ regret, and Liu et al. [2022] extend the analysis to non-stationary environments. This policy-selection approach leverages contextual bandit techniques [Dudik et al., 2011, Badanidiyuru et al., 2014] and often relies on a cost-sensitive classification oracle for efficiency. The second approach tackles BwK via a Lagrangian-dual formulation, transforming the problem into an online convex optimization (OCO) task. For example, Agrawal and Devanur [2016], Sankararaman and Slivkins [2021], Sivakumar et al. [2022], and Liu and Grigas [2022] study linear demand settings and combine bandit learning algorithms (for reward and consumption estimation [Abbasi-Yadkori et al., 2011, Auer, 2002, Sivakumar et al., 2020, Elmachtoub and Grigas, 2022, Kumar and Kleinberg, 2022, Ma et al., 2024, Zhang and Cheung, 2024]) with OCO methods to guarantee sub-linear regret. More recently, Chen et al. [2024] consider a contextual knapsack setting and show that under certain conditions one can attain constant (dimension-free) regret beyond the worst-case scenario. Our work differs by addressing continuous pricing decisions (versus discrete actions) and developing algorithms across diverse information regimes, including handling degenerate solutions that arise in resource-constrained settings.

**Online resource allocation and fluid approximations.** A related line of research examines online algorithms for resource allocation by comparing to an offline or fluid benchmark [Reiman and Wang, 2008, Jasin and Kumar, 2012, 2013, Ferreira et al., 2018, Bumpensanti and Wang, 2020,

Banerjee and Freund, 2020, Vera and Banerjee, 2021, Wang and Wang, 2022, Jiang et al., 2025, Jaillet et al., 2024, Ao et al., 2024a,b, 2025, Jiang and Zhang, 2025]. The fluid benchmark, originally introduced by Gallego and Van Ryzin [1994], provides a deterministic relaxation of the stochastic problem that serves as an upper bound on achievable revenue. For dynamic pricing with resource constraints, Wang and Wang [2022] prove that a fluid re-solving policy can achieve near-optimal $O(\log T)$ regret, but their analysis crucially assumes a *non-degeneracy condition*—requiring unique optimal dual solutions and isolated optimal bases throughout the horizon. This condition, however, often fails in practice when resources become scarce or demand patterns create degenerate solutions [Bumpensanti and Wang, 2020, Jiang et al., 2025]. Our key contribution is eliminating this restrictive requirement through boundary attraction, which preemptively reserves near-depleted resources to prevent constraint violations and maintains algorithmic stability even when the fluid solution is degenerate. Other related approaches include prophet inequality frameworks [Vera and Banerjee, 2021] and various dual-based methods [Banerjee and Freund, 2020, Ao et al., 2024a,b], but none of their techniques explicitly address the degeneracy challenge that our boundary attraction mechanism tackles in joint learning and continuous decision-making scenarios.

**Dynamic pricing with offline data.** Leveraging offline data for online pricing is increasingly important. Keskin and Zeevi [2014] pioneered the "incumbent price" setting with one accurate observation in unconstrained environments. We extend this to resource-constrained settings and provide the first rigorous analysis of how informed prices exhibit a phase transition with prediction accuracy $\varepsilon_0$. Our parameter estimation is closely related to the regression framework of Simchi-Levi and Xu [2022] and Xu and Zeevi [2020], who introduced ordinary least squares with forced exploration for contextual bandits. We extend their approach to handle hard resource constraints through three innovations: (i) periodic re-solving with estimated fluid models, (ii) rejection mechanisms for near-depleted resources, and (iii) analysis under strict feasibility requirements. Related work includes model misspecification [Ferreira et al., 2018] and offline data integration [Bu et al., 2020, Wang et al., 2024, Li et al., 2021], but without addressing resource constraints or degeneracy.

**Learning-augmented algorithms.** Our informed-price setting aligns with the emerging framework of learning-augmented algorithms [Lykouris and Vassilvitskii, 2021, Mitzenmacher and Vassilvitskii, 2022, Purohit et al., 2018, Lyu et al., 2025], which studies how to leverage potentially inaccurate predictions to improve online decisions while maintaining worst-case guarantees [Wei and Zhang, 2020]. Similar to recent work on online linear optimization with hints [Bhaskara et al., 2020, 2021, 2023], our Algorithm 3 follows the consistency-robustness paradigm: achieving near-optimal performance when predictions are accurate ($\varepsilon_0$ small) while preserving $O(\sqrt{T})$ worst-case regret guarantees when they are not. Our regret bound $O(\min\{\rho\sqrt{T}, (\varepsilon_0)^2 T + \log T\})$ exhibits the typical phase transition behavior characteristic of learning-augmented algorithms—when predictions are sufficiently accurate, the algorithm "trusts" them and achieves logarithmic regret; otherwise, it falls back to robust worst-case guarantees. However, our setting differs from standard learning-augmented frameworks in three critical ways. First, we face *hard resource constraints* requiring strict feasibility at every step, not soft constraints handled via Lagrangian relaxation. Second, we must address *degeneracy in optimal solutions* that emerges from resource scarcity, necessitating our boundary attraction mechanism. Third, resource consumption is *irreversible*—poor early decisions permanently constrain future feasibility, unlike settings where decisions can be revised. To our knowledge, we provide the first learning-augmented algorithm for resource-constrained pricing that addresses all three challenges simultaneously.

**Notation.** For a real number $x$, we use $\lceil x \rceil$ to denote the smallest integer $\geq x$ and $\lfloor x \rfloor$ for the largest integer $\leq x$. We write $x_+ = \max\{x, 0\}$. For a set $S$, let $|S|$ be its cardinality. We denote by

$$d_{\max} = \max_{p \in [L,U]^n} \|f(\boldsymbol{p})\|_2$$

the maximum $\ell_2$-norm of any feasible demand vector under the true demand function $f$.

## 2 Algorithm and Logarithmic Regret with Full Information

In the full information setting, the demand function $f(\boldsymbol{p}) = \boldsymbol{\alpha} + B\boldsymbol{p}$ is known to the decision-maker (DM). Despite this knowledge, resource constraints complicate the pricing decisions, as the DM must balance revenue maximization with resource depletion. Prior work, such as Jasin [2014], achieves logarithmic regret but relies on the non-degeneracy condition, which assumes: **Unique optimal dual solution:** The dual LP has a unique optimal vertex $\boldsymbol{\lambda}^*$ (not just a unique optimal face);

---
**Algorithm 1** Boundary Attracted Re-solve Method
---
1: **Input:** $\boldsymbol{c}^1 = \boldsymbol{C}$, $A$, $f(\boldsymbol{p}) = \boldsymbol{\alpha} + B\boldsymbol{p}$, rounding threshold $\zeta$.
2: **for** $t = 1, \ldots, T$ **do**
3:     Solve fluid model (2) for $(\boldsymbol{p}^{\pi,t}, \boldsymbol{d}^{\pi,t})$
4:     **Apply Boundary Attraction:** Set $\tilde{d}_i^t = d_i^{\pi,t}$ if $d_i^{\pi,t} \geq \zeta(T - t + 1)^{-1/2}$, else 0
5:     Set $\boldsymbol{p}^t$ s.t. $f(\boldsymbol{p}^t) = \tilde{\boldsymbol{d}}^t$
6:     Observe demand $\hat{\boldsymbol{d}}^t$, update $\boldsymbol{c}^{t+1} = \boldsymbol{c}^t - A\hat{\boldsymbol{d}}^t$
7: **end for**
---

**Isolated optimal basis:** Small perturbations in the constraint right-hand side $\boldsymbol{c}/T$ do not change which constraints are binding at optimality. More recently, Wang and Wang [2022] replaces this with assumption $\mathbf{c}^0 \neq Td^{*,T}$, (where $d^{*,T}$ is the fluid optimal demands of (1)), which means the system is either overloaded or underloaded. Moreover, its regret bound relies in order $\Omega(\|\mathbf{c}^0/T - \mathbf{d}^{*,T}\|)$ on the gap. However, degeneracy is common in practice [Bumpensanti and Wang, 2020], motivating the need for more robust algorithms.

To address this, we introduce the **Boundary Attracted Re-solve Method** (Algorithm 1), which resolves the fluid model at each step and incorporates a boundary attraction technique to handle potential degeneracy. This method adjusts the pricing strategy by setting demand to zero when it falls below a dynamic threshold, effectively pushing prices toward the upper bound $U$ and ensuring sufficient exploration of the price space.

## 2.1 Algorithm Design

At each period $t$, the DM observes the remaining capacities $\boldsymbol{c}^t$ and solves the fluid model:

$$
\begin{aligned}
V_t^{\text{Fluid}}(\boldsymbol{c}^t) = \max_{\boldsymbol{p} \in [L,U]^n} \quad & \boldsymbol{p}^\top \boldsymbol{d} \\
\text{s.t.} \quad & \boldsymbol{d} = \boldsymbol{\alpha} + B\boldsymbol{p}, \\
& A\boldsymbol{d} \leq \frac{\boldsymbol{c}^t}{T - t + 1},
\end{aligned}
\tag{2}
$$

yielding an optimal solution $(\boldsymbol{p}^{\pi,t}, \boldsymbol{d}^{\pi,t})$. To mitigate the risk of degeneracy, we define a targeted demand $\tilde{\boldsymbol{d}}^t$ by rounding small demand components to zero:

$$
\tilde{d}_i^t = \begin{cases} d_i^{\pi,t} & \text{if } d_i^{\pi,t} \geq \zeta(T - t + 1)^{-1/2}, \\ 0 & \text{otherwise}, \end{cases}
$$

where $\zeta$ is a rounding parameter. The price $\boldsymbol{p}^t$ is then set such that $f(\boldsymbol{p}^t) = \tilde{\boldsymbol{d}}^t$, ensuring that the expected demand aligns with the adjusted target. When a resource $i$ is fully depleted (i.e., $c_i^t = 0$), we reject all future demand for products requiring that resource. We leave detailed intuition on how Algorithm 1 works well in Appendix

**Why Prior Methods Fail Under Degeneracy.** When $d_j^{\pi,t} \approx 0$, the dual variable $\lambda_i$ (Lagrange multiplier for some resource $i$) can become arbitrarily large, and the gradient $\nabla^2 \mathcal{L}$ grows unboundedly as $d_j^{\pi,t} \to 0$, making the system ill-conditioned. Prior methods cannot control "moderate probability events" where small noise either exceeds remaining resources (causing irreversible infeasibility) or leads to under-allocation (wasting resources). Our boundary attraction mechanism avoids this problematic region entirely by rounding to zero, incurring a controlled total loss of

$$
\sum_{t=1}^T \|\boldsymbol{d}^{\pi,t} - \tilde{\boldsymbol{d}}_t\|_2^2 = O(\zeta^2 \log T) \quad \text{when } \zeta = O(1).
$$

This rounding cost is acceptable as it represents the price of eliminating degeneracy-related instabilities.

## 2.2 Regret Analysis

We now present the regret guarantee for Algorithm 1.

**Theorem 2.1.** *For Algorithm 1 with $\zeta \geq 4\sigma^2$, the regret is bounded by:*

$$\mathsf{Regret}^T(\pi) = O\left(\zeta^2 n^2 \|B^{-1}\|_2 \log T\right).$$

The proof, detailed in Appendix C, involves decomposing the regret using a hybrid policy that follows Algorithm 1 up to time $t$ and then optimally resolves the remaining periods without noise. By analyzing single-step differences and leveraging the boundary attraction to hedge against noise, we achieve the logarithmic regret bound without assuming non-degeneracy.

**Theoretical Improvement.** This result improves upon prior work by Jasin [2014] and Wang and Wang [2022], which require non-degeneracy for similar guarantees. Our approach avoids first-order corrections and instead uses a novel single-step difference technique inspired by Vera and Banerjee [2021], making it more robust and widely applicable.

## 3 Algorithm and Regret with No Information

In the no information setting, the demand function $f(\boldsymbol{p}) = \boldsymbol{\alpha} + B\boldsymbol{p}$ is unknown to the decision-maker (DM), who must learn it online while simultaneously making pricing decisions. This scenario presents a classic exploration-exploitation trade-off: the DM needs to gather information about the demand parameters to improve future decisions while maximizing revenue under resource constraints. Since the state-action space is continuous and high-dimensional, reinforcement learning algorithms fail in general.

To tackle this challenge, we propose an online learning algorithm that combines parameter estimation with the boundary attracted re-solve method and perturbations. Our approach ensures adequate exploration via controlled price perturbations, while leveraging the learned parameters to make near-optimal pricing decisions. We assume that we're able to reject the demands after observation.

### 3.1 Algorithm Design

At each time period $t$, the learning task can be summarized as learning the parameters $\boldsymbol{\alpha}$ and $B$. The most natural way is to adopt linear regression to learn the parameters. To be more specific, at each period $t$, given the data points $(\boldsymbol{p}^1, \boldsymbol{d}^1), \ldots, (\boldsymbol{p}^{t-1}, \boldsymbol{d}^{t-1})$, we define

$$
\begin{aligned}
D_j^t &:= \sum_{s=1}^{t-1} [d_j^s; d_j^s \cdot \boldsymbol{p}^s]^\top, \ \forall j \in [n], \\
P^t &:= \begin{bmatrix} t-1 & \sum_{s=1}^{t-1} (\boldsymbol{p}^s)^\top \\ \sum_{s=1}^{t-1} \boldsymbol{p}^s & \sum_{s=1}^{t-1} \boldsymbol{p}^s (\boldsymbol{p}^s)^\top \end{bmatrix}.
\end{aligned}
\tag{3}
$$

Then, the linear regression can be done by computing

$$
\begin{aligned}
\begin{bmatrix} \hat{\alpha}_j^t \\ \hat{\boldsymbol{\beta}}_j^t \end{bmatrix} &= (P^t)^\dagger D_j^t \\
&= \begin{bmatrix} \alpha_j \\ \boldsymbol{\beta}_j \end{bmatrix} + (P^t)^\dagger \begin{bmatrix} \sum_{s=1}^{t-1} \epsilon_j^s \\ \sum_{s=1}^{t-1} \epsilon_j^s \cdot \boldsymbol{p}^s \end{bmatrix}, \quad \forall j \in [n],
\end{aligned}
\tag{4}
$$

where $(P^t)^\dagger$ represents the pseudo-inverse of the matrix $P^t$. As we can see from (4), the estimation error (the gap between $(\hat{\boldsymbol{\alpha}}^t, \hat{B}^t)$ and $(\boldsymbol{\alpha}, B)$) scales linearly in the smallest eigenvalue of the matrix $P^t$, which can be lower bounded by the variance of the historical prices $\boldsymbol{p}^1, \ldots, \boldsymbol{p}^{t-1}$. Indeed, motivated by Keskin and Zeevi [2014], denoting $\overline{\boldsymbol{p}}^t = t^{-1} \sum_{s=1}^t \boldsymbol{p}^s$, one can show that

$$\lambda_{\min}(P^t) \tag{5}$$

$$\geq \frac{1}{n(1+2U^2)} \sum_{s=1}^{n\lfloor t/n \rfloor} (1 - \frac{1}{s}) \left\| \boldsymbol{p}^s - \overline{\boldsymbol{p}}^{s-1} - \boldsymbol{X}^{\lceil s/n \rceil} \right\|_2^2 \tag{6}$$

for some fixed anchor $\boldsymbol{X}^k, k = 0, 1, \ldots$, such that $\boldsymbol{p}^s - \overline{\boldsymbol{p}}^{s-1} - \boldsymbol{X}^k, s = kn+1, \ldots, kn+n$ form an orthogonal basis of $\mathbb{R}^n$. Therefore, in order to reduce the estimation error and guarantee sufficient exploration, instead of directly using the optimal solution of the fluid model $V^{\text{Fluid}}$, we add some

---

**Algorithm 2** Periodic-Review Re-solve with Parameter Learning

---

1: **Input:** Initial capacity $c^1 = C$, constraint matrix $A$, perturbation scale $\sigma_0$, threshold $\zeta$.
2: **for** $t = 1, \ldots, n$ **do**
3:     Sample $p^t$ uniformly from $[L, U]^n$ to initialize exploration.
4: **end for**
5: Compute initial average price $\overline{p}^n = \frac{1}{n} \sum_{t=1}^{n} p^t$.
6: **for** $t = n + 1, \ldots, T$ **do**
7:     **if** $\mod (t, n) = 1$ **then**
8:         Set block index $k = \lfloor (t - 1)/n \rfloor$.
9:         Estimate $\hat{\alpha}^{kn+1}$ and $\hat{B}^{kn+1}$ using regression on historical data.
10:        Solve the estimated fluid model with current capacity $c^{kn+1}$ to obtain $\tilde{p}^k$.
11:     **end if**
12:     Set price $p^t = \overline{p}^{t-1} + (\tilde{p}^k - \overline{p}^{kn}) + \sigma_0 t^{-1/4} e_{t-kn}$.
13:     Compute predicted demand $\tilde{d}^t = \hat{\alpha}^{kn+1} + \hat{B}^{kn+1} p^t$.
14:     Define rejection set $\mathcal{I}_r^t = \{i \in [n] : \tilde{d}_i^t \le \zeta((T - t + 1)^{-1/4} + t^{-1/4})\}$.
15:     Observe actual demand $\hat{d}^t = f(p^t) + \epsilon^t$, and reject demands in $\mathcal{I}_r^t$.
16:     Update capacity $c^{t+1} = c^t - A\hat{d}^t$.
17:     Update average price $\overline{p}^t = \frac{t-1}{t}\overline{p}^{t-1} + \frac{1}{t}p^t$.
18: **end for**

---

perturbation on the pricing strategy $p^t$ to get a higher variance. Specifically, at time $t$, we can do this by setting the desired price as

$$\tilde{p}^t = \overline{p}^{t-1} + (\tilde{p}^k - \overline{p}^{kn}) + \sigma_0 t^{-1/4} e_{t-kn}$$

for $k = \lfloor (t - 1)/n \rfloor$ and $\tilde{p}^k$ is the solution to (2) by replacing the parameters with estimated ones $\hat{\alpha}^{kn+1}, \hat{B}^{kn+1}$ and inventory level $c^{kn+1}$. As a result, the term $(\tilde{p}^k - \overline{p}^{kn})$ performs as a momemtum forward to the re-solve solution and an anchor for time steps $kn + 1, \ldots, kn + n$, hence balancing the exploitation and the exploration. More intuitions on the Algorithm design are left in Appendix A.2.

The boundary attraction mechanism, akin to the full information setting, nullifies demand for components falling below a dynamic threshold. This enhances robustness against estimation errors and stochastic noise.

### 3.2 Regret Analysis

We now provide a regret guarantee for Algorithm 2.

**Theorem 3.1.** *For Algorithm 2, with threshold $\zeta \ge C n^{5/4} \log^{3/2} n \sigma_0 \sqrt{\sigma} \log T$ for some constant $C$, the regret is bounded by:*

$$\mathsf{Regret}^T(\pi) = O\left((\zeta^2 + \|B^{-1}\|_2)\sqrt{T}\right).$$

We sketch the proof for Theorem 3.1 here, while the details are left in Appendix D. WLOG we assume $T = nT'$ for some integer $T' > 0$. The first step is similar to the proof of Theorem 2.1: we split the regret as

$$\mathsf{Regret}^T(\pi) = \mathbb{E}\left[\sum_{k=1}^{T'} \mathcal{R}^T(\mathbf{Hybrid}^k, \mathcal{F}^T) - \mathcal{R}^T(\mathbf{Hybrid}^{k+1}, \mathcal{F}^T)\right],$$

where $\mathbb{R}^T(\mathbf{Hybrid}^k, \mathcal{F}^T)$ denotes the total revenue of the hybrid policy defined as using Algorithm 2 up to time $kn$ and getting the remain revenue by solving (2) directly without noise. Now different from the proof of Theorem 2.1, we donot have the accurate parameters as well as the fluid optimal solutions. As a result, we need to give bound to the estimation error of $\hat{\alpha}^{kn+1}, \hat{B}^{kn+1}$ with respect to the true parameters $\alpha, B$, as well as the corresponding estimated solutions $p^t, f(p^t)$. To achieve this target, we briefly introduce the continuity property in strongly convex constrained optimization problem in objective function.

**Lemma 3.2** (Prop 4.32, Bonnans and Shapiro [2013]). *Suppose the constrained optimization problem* (2) *satisfies the second-order growth condition* $\boldsymbol{p}^\top \boldsymbol{d} - (\boldsymbol{p}^{\pi,t})^\top \boldsymbol{d}^{\pi,t} \geq -\kappa \mathsf{Dist}(\boldsymbol{d}, D^{\pi,t})^2$ *for some constant* $\kappa > 0$ *and feasible solution* $(\boldsymbol{p}^{\pi,t})^\top \boldsymbol{d}^{\pi,t}$ *to* (2), *where* $D^{\pi,t} = \{\boldsymbol{d}^{\pi,t} : \boldsymbol{d}^{\pi,t}, \boldsymbol{p}^{\pi,t} \text{ solve } (2)\}$. *Then we have* $\mathsf{Dist}(\hat{\boldsymbol{d}}, D^{\pi,t}) \leq C\kappa^{-1}(\left\| B - \hat{B} \right\|_2 + \|\boldsymbol{\alpha} - \hat{\boldsymbol{\alpha}}\|_2)$ *for some constant* $C > 0$, *where* $\hat{B}, \hat{\boldsymbol{\alpha}}$ *lies in some neighbor of* $B, \boldsymbol{\alpha}$ *depending on* $\lambda_{\min}(B + B^\top)$ *and* $\hat{\boldsymbol{d}}$ *is the solution to* (2) *by replacing* $\boldsymbol{\alpha}, B$ *with* $\hat{\boldsymbol{\alpha}}, \hat{B}$.

Now by applying the above lemma, we can bound the estimation error of solution $\boldsymbol{p}^t, f(\boldsymbol{p}^t)$ based on estimation error of the parameters $\hat{\boldsymbol{\alpha}}^{kn+1}, \hat{B}^{kn+1}$, which is achievable by (5). By combining the estimation error and the error caused by noises as in the proof of Theorem 2.1, we can get the final bound in Theorem 3.1.

Even in the unconstrained case, it is well-known that a $\sqrt{T}$ lower bound of worst-case regret is inevitable [Keskin and Zeevi, 2014, Chen et al., 2022]. Formally, we have

**Lemma 3.3** (Keskin and Zeevi 2014). *There exists a finite positive constant* $c > 0$ *such that* $\mathsf{Regret}^T(\pi) \geq c\sqrt{T}$ *for any online policy* $\pi$.

Therefore, we know that the regret bound presented in Theorem 3.1 is of optimal order.

# 4 Improvement with machine-learned informed price

In many practical scenarios, decision-makers leverage offline data or machine-learned models to obtain initial estimates of the demand function. We refer to this intermediate case as the *informed-price setting*, which lies between the extremes of full information and no information. Specifically, we assume access to an initial price-demand sample $(\boldsymbol{p}^0, \boldsymbol{d}^0)$, where $\boldsymbol{d}^0$ is an estimate of the true demand $f(\boldsymbol{p}^0)$, and the estimation error is bounded by a known constant $\epsilon^0$ (i.e., $\|\boldsymbol{d}^0 - f(\boldsymbol{p}^0)\|_2 \leq \epsilon^0$). This setup reflects real-world practice: firms often start with a price informed by historical data or expert knowledge and observe the resulting demand. The benefit of this informed starting point, however, hinges on the accuracy of $(\boldsymbol{p}^0, \boldsymbol{d}^0)$ and on the firm's ability to quantify the error $\epsilon^0$. This informed scenario is common in practice but has not been thoroughly studied in theory under resource constraints. In this section, we investigate how to optimally exploit the informed price when $\epsilon^0$ is known.

## 4.1 Challenges and Lower Bounds

A crucial insight is that knowing the error bound $\epsilon^0$ is necessary to improve beyond the no-information baseline. Without this knowledge, even an informed initial sample cannot guarantee better worst-case regret than $O(\sqrt{T})$. The following proposition formalizes this limitation:

**Proposition 4.1.** *There exist parameter sets* $(\alpha, B)$ *and* $(\alpha', B')$ *such that, for any policy* $\pi$ *lacking knowledge of* $\epsilon^0$, *if* $\pi$ *achieves regret* $O(T^\gamma)$ *for some* $\gamma \in (0, 1)$ *on* $(\alpha, B)$, *it incurs regret* $\Omega(T^{1-\gamma})$ *on* $(\alpha', B')$.

In other words, without a known error bound, a policy that is tuned to perform better than $\sqrt{T}$ on one demand instance will necessarily perform worse (in fact, no better than $\Omega(\sqrt{T})$) on another instance. This result implies that, absent knowledge of $\epsilon^0$, the worst-case regret remains $\Theta(\sqrt{T})$, which is the same order as in the uninformed case. Consequently, to effectively leverage the informed price, the decision-maker must know $\epsilon^0$ and incorporate this knowledge into the policy design.

## 4.2 Algorithm Design

We introduce the **Estimate-then-Select Re-solve Algorithm** (Algorithm 3), which adaptively utilizes the informed price based on the magnitude of $\epsilon^0$. The algorithm evaluates the condition $(\epsilon^0)^2 T > \rho\sqrt{T}$, where $\rho$ is a tolerance parameter. If this holds, it reverts to the no-information strategy (Algorithm 2). Otherwise, it capitalizes on the informed pair to refine parameter estimation and decision-making.

The algorithm anchors its regression on the pair $(\boldsymbol{p}^0, \boldsymbol{d}^0)$, estimating the demand parameter $B$ as follows:

$$\hat{B}^t = \left(\sum_{s=1}^{t-1}(\boldsymbol{p}^s - \boldsymbol{p}^0)(\boldsymbol{p}^s - \boldsymbol{p}^0)^\top\right)^\dagger \sum_{s=1}^{t-1}(\boldsymbol{d}^s - \boldsymbol{d}^0)(\boldsymbol{p}^s - \boldsymbol{p}^0)^\top.$$

---

**Algorithm 3** Estimate-then-Select Re-solve with Parameter Learning

---

1: **Input:** Initial capacity $c^1 = C$, matrix $A$, error bound $\epsilon^0$, tolerance $\rho$, perturbation scale $\sigma_0$, threshold $\zeta$.
2: **if** $(\epsilon^0)^2 T > \rho\sqrt{T}$ **then**
3:     Switch to Algorithm 2 (no-information setting).
4: **end if**
5: **for** $t = 1, \ldots, T$ **do**
6:     Compute $\hat{B}^t$ using regression anchored at $(\boldsymbol{p}^0, \boldsymbol{d}^0)$.
7:     Solve the estimated fluid model $\hat{V}_t^{\text{Fluid}}(\boldsymbol{c}^t)$ with $\hat{B}^t$ and $\boldsymbol{c}^t$.
8:     Perturb the solution: $\tilde{\boldsymbol{p}}^t \leftarrow \tilde{\boldsymbol{p}}^t + \sigma_0 \text{sgn}(\tilde{\boldsymbol{p}}^t - \boldsymbol{p}^0) t^{-1/4} e_{\text{ mod } (t,n)}$.
9:     Compute predicted demand $\tilde{\boldsymbol{d}}^t = \boldsymbol{d}^0 + \hat{B}^t(\tilde{\boldsymbol{p}}^t - \boldsymbol{p}^0)$.
10:     Define rejection set $\mathcal{I}_r^t = \{i \in [n] : \tilde{d}_i^t \leq \zeta((T - t + 1)^{-1/2} + t^{-1/2})\}$.
11:     Observe $\hat{\boldsymbol{d}}^t = f(\tilde{\boldsymbol{p}}^t) + \boldsymbol{\epsilon}^t$, reject demands in $\mathcal{I}_r^t$, and update $\boldsymbol{c}^{t+1} = \boldsymbol{c}^t - A\hat{\boldsymbol{d}}^t$.
12: **end for**

---

By using $\boldsymbol{p}^0$ as a reference, this approach enhances estimation precision when $\epsilon^0$ is small, thereby improving pricing decisions.

The perturbation term facilitates ongoing exploration, while a boundary attraction mechanism mitigates errors from estimation and noise, ensuring robust performance across iterations. See Appendix A.2 for more intuition.

### 4.3 Regret Analysis

The performance of Algorithm 3 is quantified in Theorem 4.2.

**Theorem 4.2.** *For Algorithm 3, the regret is bounded by:*

$$\text{Regret}^T(\pi) = O\left(\min\left\{\rho\sqrt{T}, (\epsilon^0)^2 T + C' \log T\right\}\right),$$

*where* $C' = \sigma_0 d_{\max} \|B^{-1}\|_2 + n^2 \sigma^2 \|B^{-1}\|_2 + \zeta^2$.

This bound highlights a key trade-off: when $\epsilon^0$ is small, indicating high accuracy in the informed price, the regret approaches $O(\log T)$, aligning with full-information performance. Conversely, when $\epsilon^0$ is large, the regret gracefully transitions to $O(\sqrt{T})$, matching the no-information setting. This adaptability makes the algorithm highly practical for scenarios with varying data quality.

Additionally, Proposition 4.3 confirms the tightness of this bound, showing that no algorithm can consistently achieve better worst-case regret without further assumptions.

**Proposition 4.3.** *There exist instances where any policy without knowledge of $\epsilon^0$ incurs regret* $\Omega(\max\{\rho\sqrt{T}, (\epsilon^0)^2 T\})$, *matching the upper bound in Theorem 4.2.*

Thus, Algorithm 3 optimally exploits the informed price when $\epsilon^0$ is known, effectively bridging the gap between full and no-information paradigms.

## 5 Numerical Experiments

In this section, we present numerical experiments to validate our theoretical findings and demonstrate the scalability and robustness of the proposed algorithms. We consider a simulated dynamic pricing problem with $m = 10$ resources and $n = 20$ products, and we evaluate our three algorithms (full-information, no-information, and informed-price) in various scenarios. We only present the validation of our theoretical results here, while more experiments are left in Appendix B.[1]

**Simulation setup.** We generate a random instance of the pricing problem as follows. The entries of the consumption matrix $A \in \mathbb{R}_+^{10 \times 20}$ are drawn i.i.d. from Uniform$[0, 1]$. We sample the true linear demand model by drawing $\alpha \in \mathbb{R}^{20}$ with each $\alpha_i \sim$ Uniform$[5, 10]$ and a slope matrix $B \in \mathbb{R}^{20 \times 20}$ with each $B_{ij} \sim$ Uniform$[-1, 0]$. To ensure $B$ satisfies Assumption 1.2, we subtract the largest eigenvalue of $(B + B^\top)/2$ from each diagonal entry of $B$ (this shifts the eigenvalues of the symmetric part to be negative). We then set the initial capacities $\boldsymbol{c}^0 = Ad^*$, where $d^* =$

---

[1] All the experiments in our paper are run on a Macbook pro 14 with m2 silicon chip

$\arg\max_{d \geq 0} d^\top B^{-1}(d - \alpha)$ is the optimal demand vector for the unconstrained revenue maximization problem. This choice of $c^0$ ensures that, under the optimal pricing policy, the resource constraints will be actively limiting sales (creating a non-trivial constrained problem). We introduce i.i.d. observation noise with a Gaussian distribution $\mathcal{N}(0, \sigma^2)$ and specify $\sigma_0 = 1$ for the perturbation scale in all algorithms. Unless stated otherwise, we fix the threshold parameter $\zeta = 1$ and (for the informed setting) the initial error $\epsilon^0 = 0.1$, and we vary these parameters in dedicated experiments to study their effects.

**Validation of theoretical results.** We first examine how the regret scales with the time horizon $T$, to verify the theoretical rates in each information regime. In this experiment, we set $\sigma = 1$ and choose $\epsilon^0 = T^{-1/2}$ in the informed setting (simulating a scenario where the prior estimate improves as $T$ grows). Figure 1 illustrates the regret achieved by our algorithms for $T \in \{50, 100, 200, 400, 800, 1600\}$. Each point is the average of 100 simulation runs, and the shaded region shows the 95% confidence interval. As expected, in the full-information case (Figure 1a) the regret remains nearly constant (indicating the $O(\log T)$ growth is very mild), while in the no-information case (Figure 1b) the regret grows on the order of $\sqrt{T}$. The informed-price algorithm (Figure 1c) also exhibits an almost flat regret curve, similar to the full-information case, since $\epsilon^0$ was set to scale favorably with $T$. These observations empirically confirm our theoretical guarantees for all three settings, even in scenarios where the fluid problem (2) is degenerate.

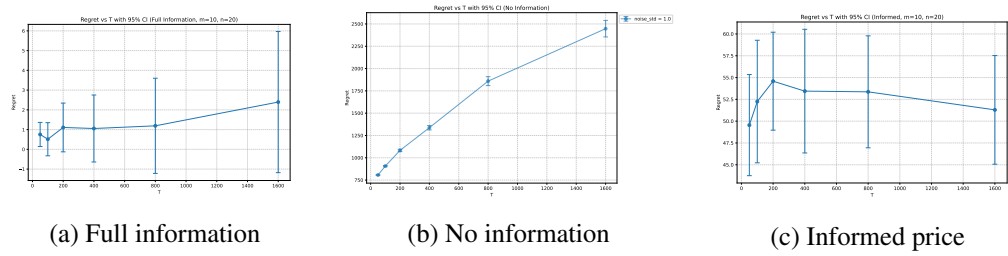

| (a) Full information | (b) No information | (c) Informed price |

Figure 1: Regret of our algorithms under different time horizons $T$ for (a) full information, (b) no information, and (c) informed price settings.

## 6 Conclusions

We have advanced the study of dynamic pricing under resource constraints by proposing three novel algorithms that effectively balance exploration and exploitation across different information regimes. Our key contributions can be summarized as follows:

1. **Boundary-Attracted Re-solve (Full Information):** We develop a pricing algorithm (Algorithm 1) for the full-information setting that achieves logarithmic regret without requiring the non-degeneracy condition assumed in prior work.

2. **Optimal Learning (No Information):** For the case with no prior demand information, we design an online learning algorithm (Algorithm 2) that attains the optimal $O(\sqrt{T})$ regret, matching the theoretical lower bound.

3. **Leveraging an Informed Price (Partial Information):** In the common situation where an initial price recommendation is available from historical data or a predictive model, we propose an estimate-then-select re-solving algorithm (Algorithm 3) that exploits this information when it is reliable. Our algorithm smoothly interpolates between full information and no information, achieving improved regret bounds when the offline estimate is accurate (and reverting to $O(\sqrt{T})$ otherwise).

Looking ahead, there are several interesting directions for future work. One direction is to extend our framework to *nonlinear* demand models, relaxing the linearity assumption while maintaining tractable regret bounds. Another direction is to consider scenarios where the error bound $\epsilon^0$ on the informed price is not given and must be estimated on the fly—developing algorithms that can learn the reliability of offline data in real time. Additionally, integrating more advanced machine learning techniques for demand forecasting could further improve performance, paving the way for more scalable and data-driven pricing strategies in complex, changing market environments.

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

# A Extended Intuitions

## A.1 Boundary Attracted Re-solve Method

Boundary attraction provides three critical advantages:

1. **Operational Safety:** Prevents noise-induced resource constraint violations by creating a buffer zone when resources near depletion. This "stop-selling-when-almost-sold-out" rule is operationally intuitive and mirrors practical inventory management;

2. **Theoretical Tractability:** Enables single-step difference analysis without requiring non-degeneracy conditions. We can bound the regret at each period independently as in Appendix C by avoiding the near-zero instability region.

3. **Robustness to Estimation:** Creates safety margin absorbing both stochastic noise (in full information) and parameter estimation errors (in learning settings).

## A.2 Variance Perturbation and Anchoring

In line 12 of Algorithm 2, we add controlled exploration noise to the price:

$$p^t = \overline{p}^{t-1} + (\tilde{p}^k - \overline{p}^{kn}) + \sigma_0 t^{-1/4} e_{t-kn} \tag{7}$$

where $e_{t-kn} \sim \mathcal{N}(0, I_n)$ is a standard Gaussian vector. This perturbation serves two purposes:

1. **Exploration**: ensures all products are priced with sufficient variation to enable accurate parameter estimation via regression.

2. **Variance control**: the $t^{-1/4}$ decay rate balances exploration early (when estimation error $\|\hat{B}^{kn+1} - B\|$ is large) against exploitation later (when estimates become accurate).

This forced exploration technique follows standard approaches in contextual bandits [Abbasi-Yadkori et al., 2011], adapted to our periodic re-solving structure.

**Anchoring regression with informed prices.** When an informed price-demand pair $(p^0, d^0)$ with known error bound $\epsilon_0$ is available, we employ an *anchoring regression* approach that leverages this prior knowledge while adapting to newly observed data. Specifically, at time $t$, we solve the constrained least squares problem:

$$\min_{\alpha, B} \quad \sum_{s=1}^{t-1} \|d^s - (\alpha + Bp^s)\|_2^2 \tag{8}$$
$$\text{subject to} \quad \|d^0 - (\alpha + Bp^0)\|_2^2 \le \epsilon_0^2.$$

The idea is closely related to the unconstrained regression in Xu and Zeevi [2020] and Simchi-Levi and Xu [2022]. Moreover, this extends naturally to multiple informed prices by solving

$$\min_{\alpha, B} \quad \sum_{s=1}^{t-1} \|d^s - (\alpha + Bp^s)\|_2^2$$
$$\text{subject to} \quad \|d^{0,i} - (\alpha + Bp^{0,i})\|_2^2 \le \epsilon_{0,i}^2, \quad \forall (p_{0,i}, d_{0,i}).$$

**Why not estimate $\epsilon_0$ online** One might wonder whether $\epsilon_0$ could be learned adaptively during the selling horizon, rather than assuming it is known. However, this approach faces fundamental challenges in our setting. Actually, testing the informed price multiple times to estimate $\epsilon_0$ permanently depletes capacity, preventing future corrections. Unlike unconstrained settings, exploration here has permanent costs. Such payoff, as suggested in Proposition 4.3, may lead to worse performance than using algorithms without the informed prices.

# B Additional Experiments

**Phase transition with respect to error bound.** Next, we investigate the effect of the prior error $\epsilon^0$ on the performance of the informed-price algorithm. In Figure 2, we plot the regret of Algorithm 3 (with tolerance $\rho = 0.1$) as a function of $T$ for different values of the misspecification error $\epsilon^0$. We observe a clear phase transition: when $\epsilon^0$ is very small, the regret grows in an $O((\epsilon^0)^2 T)$ fashion (nearly constant in this plot, since $(\epsilon^0)^2 T$ is kept low), but as $\epsilon^0$ increases beyond a threshold,

the regret growth shifts to the $O(\sqrt{T})$ regime. This behavior is consistent with Theorem 4.2 and Proposition 4.3, and it quantifies how the value of the informed price degrades as the prior becomes less accurate.

**Robustness to demand noise.** We evaluate the robustness of our algorithms under different levels of demand noise. Figure 3 shows the regret in each information setting for noise standard deviation $\sigma \in \{0.1, 0.2, 0.5, 1, 2, 5\}$ (keeping $\zeta = 1$ and $\epsilon^0 = 0.1$ fixed). The performance of all three algorithms remains relatively stable even as the noise increases by orders of magnitude. In particular, comparing the cases $\sigma = 0.1$ and $\sigma = 5$, we see only a modest increase in regret. This suggests that our methods degrade gracefully in the presence of larger demand uncertainty, underscoring their practical robustness.

**Effect of threshold parameter $\zeta$.** Finally, we study how the choice of the threshold parameter $\zeta$ (used in the boundary-attraction step of our algorithms) impacts performance. Figure 4 plots the regret for $\zeta \in \{0, 1, 2, 5, 10\}$, with $\sigma = 1$ and $\epsilon^0 = 0.1$ fixed. Moreover, when $\zeta = 0$, it serves as the classical re-solve **baseline** algorithm. We find that setting $\zeta = 0$ (i.e. no demand thresholding) significantly worsens performance, likely because the algorithms can get stuck at degenerate solutions when small-demand products are never filtered out. On the other hand, an overly large $\zeta$ (e.g. 10) also leads to higher regret, since aggressively filtering out demand can hurt revenue. A moderate value of $\zeta$ achieves the lowest regret. These results validate the importance of the boundary-attraction mechanism and indicate that tuning $\zeta$ within a reasonable range is important for best performance.

In summary, our simulation results corroborate the theoretical regret bounds in all three information settings and demonstrate the scalability and resilience of the proposed algorithms. The methods perform well with multiple resources and complex demand interactions, and they maintain strong performance even under substantial noise and model misspecification. Furthermore, the experiments highlight how incorporating a good prior (small $\epsilon^0$) and appropriately tuning parameters like $\zeta$ can yield significant practical gains.

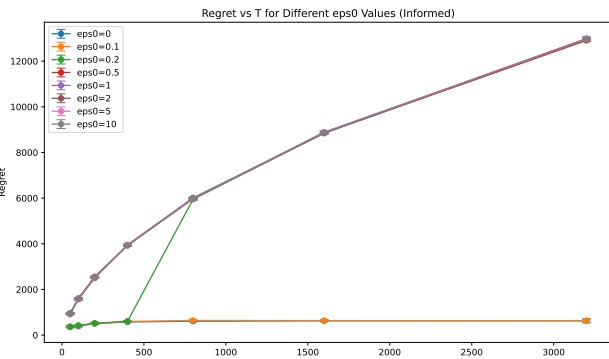

Figure 2: Regret of our algorithms under different time horizons $T$ and misspecification error $\epsilon^0$.

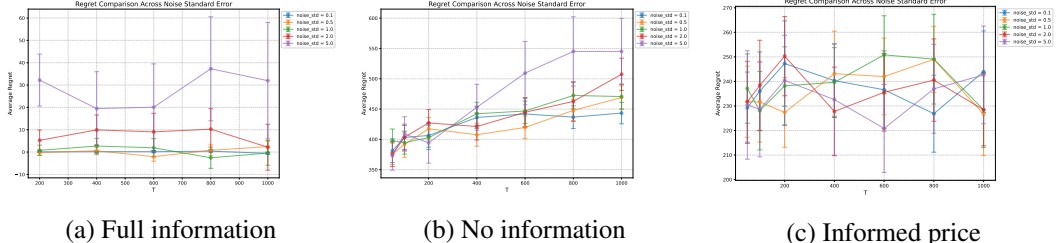

(a) Full information      (b) No information      (c) Informed price

Figure 3: Regret of our algorithms under different noise scales for (a) full information, (b) no information, and (c) informed price settings.

**Large-Scale Experimental Validation** To demonstrate the scalability and practical applicability of our algorithms, we conducted comprehensive experiments at large-scale problem sizes. We consider

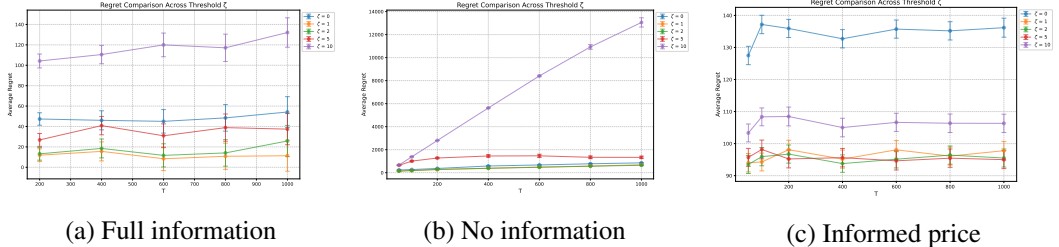

| (a) Full information | (b) No information | (c) Informed price |

Figure 4: Regret of our algorithms under different thresholding parameters $\zeta$ for (a) full information, (b) no information, and (c) informed price settings.

instances with $m = 100$ resources and $n = 200$ products, which align with the scale of real-world dynamic pricing systems in online advertising, cloud computing, and e-commerce platforms.

We generate random problem instances following the procedure described in Section 5, scaled to $m = 100$ and $n = 200$. The consumption matrix $A \in \mathbb{R}_+^{100 \times 200}$ has entries drawn i.i.d. from Uniform$[0, 1]$. The demand model parameters are $\alpha_i \sim$ Uniform$[5, 10]$ and $B_{ij} \sim$ Uniform$[-1, 0]$, with $B$ adjusted to satisfy Assumption 1.2. Initial capacities are set to $c^1 = Ad^*$ where $d^*$ is the unconstrained optimal demand, ensuring that resource constraints are binding. We fix noise standard deviation $\sigma = 1$, perturbation scale $\sigma_0 = 1$, and (for the informed setting) misspecification error $\epsilon^0 = T^{-1/2}$. Each data point represents the average over 100 independent replications, with 95% confidence intervals computed via bootstrap.

Table 1 presents the regret achieved by our algorithms and the baseline across three information settings and five time horizons. The results provide compelling evidence for the effectiveness of boundary attraction.

1. **Scalability confirmed**: Our algorithms perform effectively at realistic problem dimensions, with computational overhead remaining tractable even for $m = 100$ resources and $n = 200$ products. This validates the practical applicability of our approach to real-world systems.

2. **Consistent improvement**: Boundary attraction provides substantial regret reduction across all three information settings—30% to 45% improvement in the full-information case, 32% to 39% in learning, and 27% to 45% in the informed-price setting. Notably, the benefits persist across the entire range of time horizons tested.

3. **Statistical significance**: The non-overlapping confidence intervals between our method and the baseline confirm that all improvements are statistically significant at the 95% confidence level. This robustness is critical for deployment in production systems where reliability matters.

4. **Enhanced stability**: Our method consistently exhibits tighter confidence intervals than the baseline (e.g., $\pm 201$ vs. $\pm 655$ for full information at $T = 1000$), indicating substantially more stable performance across different demand realizations. This reduced variance stems from the boundary attraction mechanism preventing catastrophic constraint violations that cause large swings in regret.

5. **Growing long-term benefit**: The absolute performance gap widens as the time horizon increases. For instance, in the learning setting, the gap grows from $4,513$ at $T = 200$ to $35,415$ at $T = 1000$. This demonstrates that boundary attraction provides compounding benefits over extended planning horizons—a crucial property for practical applications operating over days, weeks, or months.

6. **Effectiveness across information regimes**: The boundary attraction mechanism delivers value in all three settings, confirming that degeneracy is a fundamental challenge independent of whether demand parameters are known, learned online, or partially informed by predictions. This universality underscores the importance of our contribution.

These large-scale experiments provide strong empirical validation of our theoretical contributions, demonstrating that the boundary attraction mechanism successfully addresses the degeneracy challenge at scales relevant to real-world applications.

Table 1: Large-scale experimental results: $m = 100$ resources, $n = 200$ products. Mean regret $\pm$ 95% confidence interval over 100 replications.

| Scenario | $T$ | Baseline ($\zeta = 0$) | Our Method ($\zeta = 1$) |
|---|---|---|---|
| **Full Information** | 200 | $5{,}173 \pm 259$ | $\mathbf{3{,}582 \pm 107}$ |
| | 400 | $7{,}646 \pm 382$ | $\mathbf{4{,}554 \pm 137}$ |
| | 600 | $9{,}363 \pm 468$ | $\mathbf{5{,}569 \pm 167}$ |
| | 800 | $11{,}454 \pm 573$ | $\mathbf{6{,}234 \pm 187}$ |
| | 1,000 | $13{,}100 \pm 655$ | $\mathbf{6{,}689 \pm 201}$ |
| **Learning** | 200 | $29{,}548 \pm 1{,}477$ | $\mathbf{25{,}035 \pm 751}$ |
| | 400 | $50{,}282 \pm 2{,}514$ | $\mathbf{36{,}327 \pm 1{,}090}$ |
| | 600 | $66{,}223 \pm 3{,}311$ | $\mathbf{45{,}081 \pm 1{,}352}$ |
| | 800 | $76{,}251 \pm 3{,}813$ | $\mathbf{50{,}855 \pm 1{,}526}$ |
| | 1,000 | $91{,}578 \pm 4{,}579$ | $\mathbf{56{,}163 \pm 1{,}685}$ |
| **Informed** | 200 | $25{,}661 \pm 1{,}283$ | $\mathbf{18{,}585 \pm 558}$ |
| | 400 | $38{,}288 \pm 1{,}914$ | $\mathbf{25{,}001 \pm 750}$ |
| | 600 | $50{,}748 \pm 2{,}537$ | $\mathbf{30{,}868 \pm 926}$ |
| | 800 | $64{,}406 \pm 3{,}220$ | $\mathbf{34{,}756 \pm 1{,}043}$ |
| | 1,000 | $71{,}133 \pm 3{,}557$ | $\mathbf{39{,}246 \pm 1{,}177}$ |

## C  Proof of Theorem 2.1

**Proof Roadmap**   We prove the $O(\zeta^2 n^2 \|B^{-1}\|_2 \log T)$ regret bound through the following steps:

1. **Hybrid Policy Construction:** Define a sequence of hybrid policies $\{\text{Hybrid}^t\}_{t=1}^{T+1}$ that interpolate between the offline optimal solution and our online algorithm, enabling a telescoping decomposition of regret.

2. **Single-Step Difference Analysis:** Decompose the total regret into a sum of single-step differences $\mathcal{R}^T(\text{Hybrid}^t, \mathcal{F}^T) - \mathcal{R}^T(\text{Hybrid}^{t+1}, \mathcal{F}^T)$ and analyze each term independently. Our hybrid policy decomposition is inspired by the compensated coupling technique of Vera and Banerjee [2021], but differs in two key aspects: (1) we handle continuous pricing decisions rather than discrete accept/reject choices, where the most critical "correct / incorrect decision" analysis in their setting no longer holds; (2) our novel boundary attraction mechanism explicitly prevents degeneracy, whereas other previous dynamics pricing paper assumes non-degeneracy throughout.

3. **Case-by-Case Bounding via Boundary Attraction:** For each time period, classify into three cases based on the magnitude of estimated demands relative to the rounding threshold $\zeta$:

   - **Case I**: All demands large (no rounding) — control revenue loss via noise concentration

   - **Case II**: All demands small (full rounding) — show rounding cost is negligible

   - **Case III**: Mixed demands — combine techniques from Cases I and II

4. **Concentration Inequalities:** Apply sub-Gaussian tail bounds to control the probability that noise causes constraint violations or significant deviations from the fluid benchmark.

5. **Summation and Final Bound:** Sum the per-period bounds over all $T$ periods, showing that the dominant term scales as $O(\zeta^2 \log T)$ from Case II contributions, while Cases I and III contribute lower-order terms.

**Key Innovation:** Unlike prior work requiring non-degeneracy assumptions, our boundary attraction mechanism (rounding small demands to zero) prevents the algorithm from entering degenerate regions while maintaining logarithmic regret. This is formalized through careful analysis of the three cases above.

To begin with, we restate the re-solve constrained programming problem:

$$
\begin{aligned}
\max_{p \in \mathcal{P}} \quad & r = \boldsymbol{p}^\top \boldsymbol{d} \\
\text{s.t.} \quad & \boldsymbol{d} = \boldsymbol{\alpha} + B\boldsymbol{p}, \\
& Ad \le \frac{\boldsymbol{c}^t}{T - t + 1},
\end{aligned}
\tag{9}
$$

where $\boldsymbol{c}^t$ is the inventory level at the beginning of time $t$. We denote $(p^{\pi,t}, \boldsymbol{d}^{\pi,t})$ as the optimal solution to (9). For notational convenience, we use $p(\boldsymbol{d}) := B^{-1}(\boldsymbol{d} - \boldsymbol{\alpha}), r(\boldsymbol{d}) := p(\boldsymbol{d})^\top \boldsymbol{d}$ to represent the unnoised maps of demand to price and demand to revenue, respectively. Moreover, we use $r(\boldsymbol{d}, \boldsymbol{\epsilon}) := r(\boldsymbol{d}) + p(\boldsymbol{d})^\top \boldsymbol{\epsilon}$ to represent the revenue with noised demand. Throughout the proof, we use $\pi$ to denote our online policy given in Algorithm 1. Given filtration $\mathcal{F}^T$ and policy $\pi'$, we use $\mathcal{R}^T(\pi', \mathcal{F}^T)$ to represent the total revenue under policy $\pi'$ and realized sample path $\mathcal{F}^T$. Now we introduce the following concept of **Hybrid** policy, which is crucial in our proof.

**Definition C.1.** For $1 \le t \le T+1$, we define **Hybrid**$^t$ as the policy that applies online policy $\pi$ in time $1, \ldots, t-1$ and $\boldsymbol{d}^{\pi,t}$, while for the periods $[t+1, T]$, there are no noises of demand. Moreover, define **Hybrid**$^1$ as the fluid optimal policy given in (??) without noises and **Hybrid**$^{T+1} = \pi$ as the online policy $\pi$ throughout the process.

By definition, we have

$$
\mathcal{R}^T(\textbf{Hybrid}^t, \mathcal{F}^T) = \sum_{s=1}^{t-1} r(\boldsymbol{d}^{\pi,s}, \boldsymbol{\epsilon}^s) + (T - t + 1) r(\boldsymbol{d}^{\pi,t}).
$$

Therefore, it holds that $\mathbb{E}\left[\mathcal{R}^T(\textbf{Hybrid}^t, \mathcal{F}^T)\right] \ge \mathbb{E}\left[f(\textbf{Hybrid}^{t+1}, \mathcal{F}^T)\right], 0 \le t \le T-1$ by the convexity of (9). The regret can then be decomposed as follows:

$$
\begin{aligned}
\text{Regret}^T(\pi) &= \mathbb{E}\left[\sum_{t=1}^T \mathcal{R}^T(\textbf{Hybrid}^t, \mathcal{F}^T) - \mathcal{R}^T(\textbf{Hybrid}^{t+1}, \mathcal{F}^T)\right] \\
&= \sum_{t=1}^T \mathbb{E}\left[\mathcal{R}^T(\textbf{Hybrid}^t, \mathcal{F}^T) - \mathcal{R}^T(\textbf{Hybrid}^{t+1}, \mathcal{F}^T)\right].
\end{aligned}
\tag{10}
$$

We now focus on how to give bound to the term $\mathbb{E}\left[\mathcal{R}^T(\textbf{Hybrid}^t, \mathcal{F}^T) - \mathcal{R}^T(\textbf{Hybrid}^{t+1}, \mathcal{F}^T)\right]$ for $0 \le t \le T-1$.

By definition, the realized demand at time $t$ is given by $\boldsymbol{d}^t = \boldsymbol{d}^{\pi,t} + \boldsymbol{\epsilon}^t$ and we have

$$
\begin{aligned}
r(\boldsymbol{d}^{\pi,t}, \boldsymbol{\epsilon}^t) &= r(\boldsymbol{d}^{\pi,t}) + (\boldsymbol{\epsilon}^t)^\top \boldsymbol{p}^{\pi,t}, \\
r(d') &= r(d) + (d' - d)^\top B^{-1}(d' - d) + (2d - \boldsymbol{\alpha})^\top B^{-1}(d' - d), \quad \forall \boldsymbol{d}, d' \in \mathbb{R}_+^n
\end{aligned}
\tag{11}
$$

The single-step difference can then be rewritten as:

$$
\mathcal{R}^T(\textbf{Hybrid}^t, \mathcal{F}^T) - \mathcal{R}^T(\textbf{Hybrid}^{t+1}, \mathcal{F}^T) = (T - t + 1) r(\boldsymbol{d}^{\pi,t}) - r(\boldsymbol{d}^{\pi,t}, \boldsymbol{\epsilon}^t) - (T - t) r(\boldsymbol{d}^{\pi,t+1})
\tag{12}
$$

Now we proceed with the following three cases: case (I) $\min_i d_i^{\pi,t} \ge \zeta(T - t + 1)^{-1/2}$; case (II) $\max_i \boldsymbol{d}^{\pi,t} \le \zeta(T - t + 1)^{-1/2}$; (III) $\min_i d_i^{\pi,t} \le \zeta(T - t + 1)^{-1/2} \le \max_i d_i^{\pi,t}$.

**Case (I)** $\min_i d_i^{\pi,t} > \zeta(T - t + 1)^{-1/2}$. For convenience, we omit the notation for conditional expectation / probability of the event $\{\boldsymbol{d}^{\pi,t} > \zeta(T - t + 1)^{-1/2}\}$. Denote $\mathcal{E}_i^t$ as the event $d_i^t \ge \frac{\epsilon_i^t}{T-t}, \forall i \in [n]$ and $\mathcal{E}^t = \cap_{i=1}^n \mathcal{E}_i^t$. We now split the (12) according to

$$
\begin{aligned}
&\mathbb{E}\left[\mathcal{R}^T(\textbf{Hybrid}^t, \mathcal{F}^T) - \mathcal{R}^T(\textbf{Hybrid}^{t+1}, \mathcal{F}^T) \big| \mathcal{F}^{t-1}\right] \\
&= \mathbb{P}(\mathcal{E}^t) \mathbb{E}\left[\mathcal{R}^T(\textbf{Hybrid}^t, \mathcal{F}^T) - \mathcal{R}^T(\textbf{Hybrid}^{t+1}, \mathcal{F}^T) \big| \mathcal{E}^t, \mathcal{F}^t\right] \\
&\quad + \mathbb{P}((\mathcal{E}^t)^c) \mathbb{E}\left[\mathcal{R}^T(\textbf{Hybrid}^t, \mathcal{F}^T) - \mathcal{R}^T(\textbf{Hybrid}^{t+1}, \mathcal{F}^T) \big| (\mathcal{E}^t)^c, \mathcal{F}^{t-1}\right].
\end{aligned}
\tag{13}
$$

For the first term in (13), condition on $\mathcal{E}^t$, by definition we have

$$
\begin{aligned}
&\mathbb{E}\left[\mathcal{R}^T(\mathbf{Hybrid}^t, \mathcal{F}^T) - \mathcal{R}^T(\mathbf{Hybrid}^{t+1}, \mathcal{F}^T)\big|\mathcal{E}^t, \mathcal{F}^{t-1}\right] \\
&= \mathbb{E}\left[(T - t + 1)r(\boldsymbol{d}^{\pi,t}) - r(\boldsymbol{d}^{\pi,t}, \boldsymbol{\epsilon}^t) - (T - t)r(\boldsymbol{d}^{\pi,t+1})\big|\mathcal{E}^t, \mathcal{F}^{t-1}\right] \\
&\leq \mathbb{E}\left[(T - t + 1)r(\boldsymbol{d}^{\pi,t}) - r(\boldsymbol{d}^{\pi,t}, \boldsymbol{\epsilon}^t) - (T - t)r(\boldsymbol{d}^{\pi,t} - \frac{\boldsymbol{\epsilon}^t}{T-t})\big|\mathcal{E}^t, \mathcal{F}^{t-1}\right] \\
&\overset{(a)}{=} \mathbb{E}\left[-(\boldsymbol{\epsilon}^t)^\top \boldsymbol{p}^{\pi,t} + (T-t)\frac{(\boldsymbol{\epsilon}^t)^\top}{T-t}B^{-1}\frac{\boldsymbol{\epsilon}^t}{T-t} + (T-t)(2\boldsymbol{d}^{\pi,t} - \boldsymbol{\alpha})^\top B^{-1}\frac{\boldsymbol{\epsilon}^t}{T-t}\big|\mathcal{E}^t, \mathcal{F}^{t-1}\right] \\
&\overset{(b)}{=} \mathbb{E}\left[(\boldsymbol{d}^{\pi,t})^\top B^{-1}\boldsymbol{\epsilon}^t + \frac{1}{T-t}(\boldsymbol{\epsilon}^t)^\top B^{-1}\boldsymbol{\epsilon}^t\big|\mathcal{E}^t, \mathcal{F}^{t-1}\right],
\end{aligned}
$$

$$(14)$$

where $(a)$ follows from a similar argument in (11) and $(b)$ follows from the definition: $p^{\pi,t} = B^{-1}(\boldsymbol{d}^{\pi,t} - \boldsymbol{\alpha})$. We then have

$$
\begin{aligned}
&\mathbb{P}(\mathcal{E}^t)\mathbb{E}\left[(\boldsymbol{d}^{\pi,t})^\top B^{-1}\boldsymbol{\epsilon}^t|\mathcal{E}^t, \mathcal{F}^{t-1}\right] \\
&= \mathbb{E}\left[(\boldsymbol{d}^{\pi,t})^\top B^{-1}\boldsymbol{\epsilon}^t\right] - \mathbb{P}((\mathcal{E}^t)^c)\mathbb{E}\left[(\boldsymbol{d}^{\pi,t})^\top B^{-1}\boldsymbol{\epsilon}^t|(\mathcal{E}^t)^c, \mathcal{F}^{t-1}\right] \\
&= -\mathbb{E}\left[(\boldsymbol{d}^{\pi,t})^\top B^{-1}\boldsymbol{\epsilon}^t \mathbb{1}\{(\mathcal{E})^c\}|\mathcal{F}^{t-1}\right] \\
&\overset{(a)}{\leq} \left\|B^{-1}\right\|_2 d_{\max}\mathbb{P}((\mathcal{E})^c|\mathcal{F}^{t-1})^{1/2}\mathbb{E}\left[\left\|\boldsymbol{\epsilon}^t\right\|_2^2|\mathcal{F}^{t-1}\right]^2 \\
&= \left\|B^{-1}\right\|_2 d_{\max}\mathbb{P}((\mathcal{E})^c|\mathcal{F}^{t-1})^{1/2}\mathbb{E}\left[\left\|\boldsymbol{\epsilon}^t\right\|_2^2\right]^{1/2}
\end{aligned}
$$

$$(15)$$

where we applied Cauchy-Schwarz inequality in (a). Similarly, we have

$$
\mathbb{P}(\mathcal{E}^t)\mathbb{E}\left[-\frac{(\boldsymbol{\epsilon}^t)^\top B^{-1}\boldsymbol{\epsilon}^t}{T-t}\big|\mathcal{E}^t, \mathcal{F}^{t-1}\right] \leq \frac{\left\|B^{-1}\right\|_2}{T-t}\mathbb{E}\left[(\boldsymbol{\epsilon}^t)^2\right]. \tag{16}
$$

On the other hand, with a similar argument, we can give bound to the term

$$
\begin{aligned}
&\mathbb{P}((\mathcal{E}^t)^c)\mathbb{E}\left[f(\mathbf{Hybrid}^{t-1}) - f(\mathbf{Hybrid}^t)\big|(\mathcal{E}^t)^c\right] \\
&\leq \mathbb{P}((\mathcal{E}^t)^c)\mathbb{E}\left[(T-t)r(\boldsymbol{d}^{\pi,d})t)r(\boldsymbol{d}^{\pi,t}) - (\boldsymbol{\epsilon}^t)^\top \boldsymbol{p}^{\pi,t}|(\mathcal{E}^t)^c\right] \\
&\leq \mathbb{P}((\mathcal{E}^t)^c)\left((T-t)r_{\max} - \mathbb{E}\left[(\boldsymbol{\epsilon}^t)^\top \boldsymbol{p}^{\pi,t}|(\mathcal{E}^t)^c\right]\right) \\
&\leq (T-t)r_{\max}\mathbb{P}((\mathcal{E}^t)^c) + \sqrt{n}U\mathbb{E}\left[(\boldsymbol{\epsilon}^t)^2\right]^{1/2}\mathbb{P}((\mathcal{E}^t)^c)^{1/2},
\end{aligned}
$$

$$(17)$$

where we again use Cauchy-Schwarz inequality in the last line.

Now combining (13) (14) (15) (16) (17), we get

$$
\begin{aligned}
&\mathbb{E}\left[\mathcal{R}^T(\mathbf{Hybrid}^t, \mathcal{F}^T) - \mathcal{R}^T(\mathbf{Hybrid}^{t+1}, \mathcal{F}^T)\right] \\
&\leq \sigma(\left\|B^{-1}\right\|_2 d_{\max} + \sqrt{n}U)\mathbb{P}((\mathcal{E}^t)^c)^{1/2} + (T-t)r_{\max}\mathbb{P}((\mathcal{E}^t)^c) + \frac{\sigma^2\left\|B^{-1}\right\|_2}{T-t}, \tag{18}
\end{aligned}
$$

Now in order to give bound to $\mathbb{P}((\mathcal{E}^t)^c)$, we introduce the following concentration inequality. This standard concentration inequality controls the probability that the maximum of sub-Gaussian random variables exceeds a threshold. We use it to bound the probability that demand noise $\epsilon^t$ violates resource constraints, ensuring that boundary attraction successfully prevents infeasibility with high probability.

**Lemma C.2** (Wainwright 2019). *Let $X_1, \ldots, X_n$ be $\sigma^2$-sub-Gaussian random variables with zero mean, then for each $\lambda > 0$, it holds that*

$$
\mathbb{P}\left(\max_{1 \leq i \leq n} X_i \geq \lambda\right) \leq n\exp(-\lambda/2\sigma^2)
$$

By Lemma D.1, we can bound the term $\mathbb{P}((\mathcal{E}^t)^c)$ as

$$
\begin{aligned}
\mathbb{P}((\mathcal{E}^t)^c) &= \mathbb{P}(\exists i \in [n], \text{s.t.} d_i^{\pi,[t,T]} < \frac{\epsilon_i^t}{T-t}) \\
&\leq \mathbb{P}(\frac{\zeta}{(T-t+1)^{1/2}} < \max_i \frac{\epsilon_i^t}{T-t}) \\
&\overset{(a)}{\leq} n \exp(-2(T-t)\log n) \\
&= \exp(-2(T-t)),
\end{aligned}
\tag{19}
$$

where (a) is detived from Lemma D.1 and the fact that $\min_i d_i^{\pi,t} \geq \zeta(T-t+1)^{-1/2}$ and $\zeta \geq 4\sigma^2 \log n$. Plugging the above inequality into (18) leads to

$$
\mathbb{E}\left[\mathcal{R}^T(\mathbf{Hybrid}^t, \mathcal{F}^T) - \mathcal{R}^T(\mathbf{Hybrid}^{t+1}, \mathcal{F}^T)\right]
$$

$$
\leq \sigma(\left\|B^{-1}\right\|_2 d_{\max} + \sqrt{n}U)\exp(-(T-t)) + r_{\max}(T-t)\exp(-2(T-t)) + \frac{\sigma^2 \left\|B^{-1}\right\|_2}{T-t}.
\tag{20}
$$

**Case (II):** $\max_i d_i^{\pi,} \leq \zeta/(T-t+1)^{-1/2}$. In this case, we have $\boldsymbol{d}^{\pi,[t,T]} = \boldsymbol{\epsilon}^t = 0$. Using (12), the single-step difference in case (II) can be upper bounded by

$$
\mathbb{E}\left[\mathcal{R}^T(\mathbf{Hybrid}^t, \mathcal{F}^T) - \mathcal{R}^T(\mathbf{Hybrid}^{t+1}, \mathcal{F}^T)\right]
$$

$$
\leq \mathbb{E}\left[(T-t+1)r(\boldsymbol{d}^{\pi,t}) - (T-t)r\left(\frac{T-t+1}{T-t}\boldsymbol{d}^{\pi,t}\right)\right]
$$

$$
= \mathbb{E}\left[(T-t+1)(\boldsymbol{d}^{\pi,t})^\top B^{-1}(\boldsymbol{d}^{\pi,t} - \boldsymbol{\alpha}) - (T-t)\frac{T-t+1}{T-t}(\boldsymbol{d}^{\pi,t})^\top B^{-1}(\frac{T-t+1}{T-t}\boldsymbol{d}^{\pi,t} - \boldsymbol{\alpha})\right]
$$

$$
= \mathbb{E}\left[-\frac{T-t+1}{T-t}(\boldsymbol{d}^{\pi,t})^\top B^{-1}\boldsymbol{d}^{\pi,t}\right]
$$

$$
\leq \frac{2n^2\zeta^2 \left\|B^{-1}\right\|_2}{T-t+1}
\tag{21}
$$

**Case (III):** $\min_i d_i^{\pi,t} < \zeta(T-t+1)^{-1/2} < \max_i d_i^{\pi,t}$. In this case, we can derive upper bound by following both Case (I) and Case (II). To be more specific, let $\mathcal{I} = \{i \in [n] : d_i^{\pi,t} > \zeta(T-t+1)^{-1/2}\}$, $\overline{\mathcal{I}} = [n]\backslash\mathcal{I}$ and $\mathcal{E}_{\mathcal{I}} = \cap_{i \in \mathcal{I}}\mathcal{E}_i$. Let $\epsilon_{\mathcal{I}}^t$ be the vector that has components $\epsilon_i^t$ for $i \in \mathcal{I}$ and 0 otherwise. We let $\tilde{\boldsymbol{d}}^t$ be vector with components $d_i^{\pi,t} - \boldsymbol{\epsilon}^t/(T-t), i \in \mathcal{I}$ and $(T-t+1)d_i^{\pi,t}/(T-t), i \in \overline{\mathcal{I}}$. Then the single-step difference can be upper bounded by

$$
\mathbb{E}\left[\mathcal{R}^T(\mathbf{Hybrid}^t, \mathcal{F}^T - \mathcal{R}^T(\mathbf{Hybrid}^{t+1}, \mathcal{F}^T)\right]
$$

$$
\leq \mathbb{E}\left[(T-t+1)r(\boldsymbol{d}^{\pi,t}) - r(\boldsymbol{d}^t, \boldsymbol{\epsilon}^t) - (T-t)r(\tilde{\boldsymbol{d}}^t)\right]
$$

$$
= \mathbb{E}\left[(p^t)^\top\boldsymbol{\epsilon}^t - (\boldsymbol{d}^t - \boldsymbol{d}^{\pi,t})^\top B^{-1}(\boldsymbol{d}^t - \boldsymbol{d}^{\pi,t}) - (2\boldsymbol{d}^{t,\pi} - \boldsymbol{\alpha})^\top B^{-1}(\boldsymbol{d}^t - \boldsymbol{d}^{\pi,t})\right]
$$

$$
- (T-t)\mathbb{E}\left[(\tilde{\boldsymbol{d}}^t - \boldsymbol{d}^{\pi,t})^\top B^{-1}(\tilde{\boldsymbol{d}}^t - \boldsymbol{d}^{\pi,t}) + (2\boldsymbol{d}^{\pi,t} - \boldsymbol{\alpha})^\top B^{-1}(\tilde{\boldsymbol{d}}^t - \boldsymbol{d}^{\pi,t})\right].
\tag{22}
$$

Note that $d_i^t = d_i^{\pi,t}$ for $i \in \mathcal{I}$ and $d_i^t = 0$ for $i \in \overline{\mathcal{I}}$. Moreover, $d_i^{\pi,t} \leq \zeta(T-t+1)^{-1/2}, i \in \overline{\mathcal{I}}$. We have

$$
(\boldsymbol{d}^t - \boldsymbol{d}^{\pi,t})^\top B^{-1}(\boldsymbol{d}^t - \boldsymbol{d}^{\pi,t}) \leq \zeta^2(n - |\mathcal{I}|)\left\|B^{-1}\right\|_2(T-t+1)^{-1}.
\tag{23}
$$

On the other hand, we have

$$
(\tilde{\boldsymbol{d}}^t - \boldsymbol{d}^{\pi,t})^\top B^{-1}(\tilde{\boldsymbol{d}}^t - \boldsymbol{d}^{\pi,t})
$$

$$
= \frac{1}{(T-t)^2}(\boldsymbol{d}_{\overline{\mathcal{I}}}^{\pi,t})^\top B^{-1}\boldsymbol{d}_{\overline{\mathcal{I}}}^{\pi,t} - \frac{2}{(T-t)^2}(\boldsymbol{d}_{\overline{\mathcal{I}}}^{\pi,t})^\top B^{-1}\epsilon_{\mathcal{I}}^t + \frac{1}{(T-t)^2}(\epsilon_{\mathcal{I}}^t)^\top B^{-1}\epsilon_{\mathcal{I}}^t.
\tag{24}
$$

Moreover, for the first-order terms, we have

$$
(2\boldsymbol{d}^{t,\pi} - \boldsymbol{\alpha})^\top B^{-1}(\boldsymbol{d}^t - \boldsymbol{d}^{\pi,t}) = -(2\boldsymbol{d}^{t,\pi} - \boldsymbol{\alpha})^\top B^{-1}\boldsymbol{d}_{\overline{\mathcal{I}}}^{\pi,t},
\tag{25}
$$

and

$$(2\boldsymbol{d}^{\pi,t} - \boldsymbol{\alpha})B^{-1}(\tilde{\boldsymbol{d}}^t - \boldsymbol{d}^{\pi,t}) = \frac{1}{T-t}(2\boldsymbol{d}^{\pi,t} - \boldsymbol{\alpha})^\top B^{-1}\boldsymbol{d}^{\pi,t} - \frac{1}{T-t}(2\boldsymbol{d}^{\pi,t} - \boldsymbol{\alpha})^\top B^{-1}\epsilon_{\mathcal{I}}^\top.$$
(26)

Plugging (23), (24), (25) and (26) into (22) yields as similar bound as in Case (I) and Case (II):

$$\mathbb{E}\left[\mathcal{R}^T(\textbf{Hybrid}^t, \mathcal{F}^T) - \mathcal{R}^T(\textbf{Hybrid}^{t+1}, \mathcal{F}^T)\right]$$

$$\leq \sigma(\left\|B^{-1}\right\|_2 d_{\max} + \sqrt{n}U)\exp(-(T-t)) + r_{\max}(T-t)\exp(-2(T-t)) + \frac{2n^2\zeta^2 \left\|B^{-1}\right\|_2}{T-t+1}.$$
(27)

**Wrap-up.**  Combining (20), (21) and (27) leads to

$$\mathbb{E}\left[\mathcal{R}^T(\textbf{Hybrid}^t, \mathcal{F}^T) - \mathcal{R}^T(\textbf{Hybrid}^{t+1}, \mathcal{F}^T)\right]$$
$$\leq \frac{2n^2\zeta^2 \left\|B^{-1}\right\|_2}{T-t+1} + C_0\exp(-(T-t)) + \frac{\sigma^2 \left\|B^{-1}\right\|_2}{T-t},$$

where $C_0 = C'\sigma(\left\|B^{-1}\right\|_2 d_{\max} + \sqrt{n}U) + r_{\max}$ for some absolute constant $C'$. Plugging them into (10) yields

$$\text{Regret}^T(\pi) = \sum_{t=1}^{T}\mathbb{E}\left[\mathcal{R}^T(\textbf{Hybrid}^t, \mathcal{F}^T) - \mathcal{R}^T(\textbf{Hybrid}^{t+1}, \mathcal{F}^T)\right]$$
$$= O(2\zeta^2 n^2 \left\|B^{-1}\right\|_2 \log(T) + C_0).$$

## D   Proof of Theorem 2

**Proof Roadmap**   We establish the $O\left((\zeta^2 + \|B^{-1}\|_2)\sqrt{T}\right)$ regret bound through the following steps:

1. **Periodic Hybrid Policy Construction:** Extend the hybrid policy framework to accommodate periodic re-solve updates every $n$ periods, where parameters $\hat{\boldsymbol{\alpha}}^{kn+1}, \hat{B}^{kn+1}$ are re-estimated via linear regression.

2. **Error Decomposition:** Decompose the demand estimation error $\Delta^t = \boldsymbol{d}^t - \boldsymbol{d}^{\pi,t}$ into three components:

   - $\Delta_I^t$: Parameter estimation error (from regression on historical data)
   - $\Delta_{II}^t$: Mean price drift (from averaging over periods)
   - $\Delta_{III}^t$: Exploration perturbation (deliberate noise for sufficient data variance)

3. **Parameter Estimation Analysis:** Bound $\|\Delta_I^t\|_2$ using:

   - Fisher information lower bounds on data variance (Lemma from Keskin and Zeevi [2014])
   - Lipschitz continuity of constrained optimization solutions (Lemma D.2)
   - Second-order growth conditions (Lemma D.3)

   Establish $\mathbb{E}[\|\Delta_I^t\|_2^2] = O(n^5/\sqrt{k})$ where $k = \lfloor t/n \rfloor$.

4. **Single-Period Regret Bounds:** For each period $k \in [T']$, analyze the $n$-step regret $\mathcal{R}^T(\text{Hybrid}^{kn+1}, \mathcal{F}^T) - \mathcal{R}^T(\text{Hybrid}^{(k+1)n+1}, \mathcal{F}^T)$ through the same three-case framework as in Appendix C, but with:

   - Modified rounding thresholds: $\zeta[(T-t+1)^{-1/4} + t^{-1/4}]$ (vs. $(T-t+1)^{-1/2}$ in full info)
   - Additional terms from estimation errors $\Delta_I^t, \Delta_{II}^t, \Delta_{III}^t$

5. **Aggregation Over Periods:** Sum bounds over all $T' = T/n$ re-solve epochs. The key terms are:

- Estimation error: $\sum_{k=1}^{T'} O(1/\sqrt{k}) = O(\sqrt{T})$

- Noise accumulation: $O(\sqrt{T})$ from concentration inequalities

- Exploration cost: $O(\sigma_0\sqrt{T})$ from perturbations

**Connection to Prior Work:** Our parameter estimation technique builds on the regression framework of Keskin and Zeevi [2014], Xu and Zeevi [2020], Simchi-Levi and Xu [2022], but extends it to handle resource constraints through:

- Boundary attraction mechanism (preventing infeasibility)

- Periodic re-optimization (balancing exploration vs. exploitation)

- Careful perturbation design (ensuring sufficient data variance for accurate estimation)

The detailed proof follows below.

In this section, we follow a similar streamline in Appendix C. To begin with, recall our re-solve constrained programming problem:

$$
\begin{aligned}
\max_{p \in \mathcal{P}} \quad & r = \boldsymbol{p}^\top \boldsymbol{d} \\
\text{s.t.} \quad & \boldsymbol{d} = \boldsymbol{\alpha} + B\boldsymbol{p}, \\
& A\boldsymbol{d} \le \frac{\boldsymbol{c}^t}{T - t + 1},
\end{aligned}
\tag{28}
$$

where $\boldsymbol{c}^t$ is the inventory level at the beginning of time $t$. For time $t$ and $k = \lfloor (t-1)/n \rfloor$, we use the linear regression (4) to fit the coefficients $\hat{\boldsymbol{\alpha}}^{kn+1}, \hat{B}^{kn+1}$ and then substitute them into (28) with inventory level $\boldsymbol{c}^{kn+1}$ to calculate price $\hat{\boldsymbol{p}}^k$. WLOG we assume that $T = T'n$ for some integer $T'$. Since we're now using a periodic-review re-solve policy, we proceed by modifying the single-step difference in (10):

$$
\begin{aligned}
\text{Regret}^T(\pi) &= \mathbb{E}\left[ \sum_{k=1}^{T'} \mathcal{R}^T(\mathbf{Hybrid}^{kn+1}, \mathcal{F}^T) - \mathcal{R}^T(\mathbf{Hybrid}^{(k+1)n+1}, \mathcal{F}^T) \right] \\
&= \sum_{k=1}^{T'} \mathbb{E}\left[ \mathcal{R}^T(\mathbf{Hybrid}^{kn+1}, \mathcal{F}^T) - \mathcal{R}^T(\mathbf{Hybrid}^{(k+1)n+1}, \mathcal{F}^T) \right].
\end{aligned}
\tag{29}
$$

For brevity, we use $k, k+1$ to replace the superscript $kn+1, (k+1)n+1$, respectively when the context is clear. We now focus on how ot give bound to term $\mathbb{E}\left[ \mathcal{R}^T(\mathbf{Hybrid}^k, \mathcal{F}^T) - \mathcal{R}^T(\mathbf{Hybrid}^{k+1}, \mathcal{F}^T) \right], \forall k \in [T']$.

Recall that for $t \in [T]$ and $k = \lfloor (t-1)/n \rfloor$, $\boldsymbol{p}^t, \boldsymbol{d}^t := f(\boldsymbol{p}^t)$ are the prices and demands at time $t$ by Algorithm 2. We use $\Delta^t = \boldsymbol{d}^t - \boldsymbol{d}^{\pi,k}$ to denote the difference between the targeted demand and the accurate fluid optimal demands by solving (2) with inventory level $\boldsymbol{c}^{kn+1}$. Then $\Delta^t$ can be decomposed into two parts of errors as

$$
\Delta^t = \underbrace{(\tilde{\boldsymbol{d}}^k - \boldsymbol{d}^{\pi,nk+1})}_{:=\Delta_I^t} + \underbrace{\overline{\boldsymbol{d}}^{t-1} - \overline{\boldsymbol{d}}^{kn}}_{:=\Delta_{II}^t} + \underbrace{\sigma_0 t^{-1/4} B e_{t-kn}}_{:=\Delta_{III}^t}.
\tag{30}
$$

Here $\Delta_I^t$ is the estimation error, $\Delta_{II}^t$ is the shift of mean price and $\Delta_{III}^t$ is the perturbation error. With triangular inequality, for $\Delta_{II}^t$ we have

$$
\begin{aligned}
\left\|\Delta_{II}^t\right\|_2 &= \left\|\overline{\boldsymbol{d}}^{t-1} - \overline{\boldsymbol{d}}^{kn}\right\|_2 \\
&\leq \frac{t-1-kn}{(t-1)}\left\|\overline{\boldsymbol{d}}^{kn}\right\|_2 + \frac{1}{t-1}\left\|\sum_{s=kn+1}^{t-1}\boldsymbol{d}^s\right\|_2 \\
&\leq \frac{n}{t-1}\boldsymbol{d}_{\max} + \frac{n}{t-1}\boldsymbol{d}_{\max} \\
&= \frac{2n}{t-1}\boldsymbol{d}_{\max}.
\end{aligned}
\tag{31}
$$

For $\Delta_{III}^t$, it directly follows from the definition that

$$
\left\|\Delta_{III}^t\right\|_2 \leq \sigma_0\|B\|_2\, t^{-1/4}.
\tag{32}
$$

Now we proceed by giving bound to the term $\Delta_I^t$. We define a quantity $J^t$ that is crucial for deriving tail bound for the estimation error. Formally, we let

$$
\begin{aligned}
J^k &:= n^{-1}\sum_{s=1}^{nk}(1-s^{-1})\left\|p^s - \overline{\boldsymbol{p}}^{s-1} - (\tilde{\boldsymbol{p}}^k - \overline{\boldsymbol{p}}^{kn})\right\|_2^2 \\
&= n^{-1}\sum_{l=1}^{k}\sum_{i=1}^{n}\left(1 - \frac{1}{n(l-1)+i}\right)\sigma_0^2(kn+i)^{-1/2} \\
&\geq \frac{\sigma_0^2\sqrt{kn}}{8n}, \quad \forall t = 1,2,\ldots.
\end{aligned}
\tag{33}
$$

Now we introduce the following lemma for giving bound to the parameter error. This lemma provides exponential concentration for parameter estimation error $\|\hat{B}^{kn+1} - B\|_2$ around its mean, conditional on sufficient data variance $J^k$. The exponential decay rate depends on both the estimation error magnitude $\lambda$ and the accumulated information $J^k$, ensuring that with enough exploration (large $J^k$), our parameter estimates become accurate with high probability.

**Lemma D.1** (Keskin and Zeevi 2014). *Under the our choice of $p^t$, there exists constant $C_1,\sigma_1$ such that*

$$
\mathbb{P}\left(\left\|\hat{\boldsymbol{\alpha}}^{kn+1} - \boldsymbol{\alpha}\right\|_2 + \left\|\hat{B}^{kn+1} - B\right\|_2 > \lambda, J^k \geq \lambda'\right) \leq C_1(kn)^{n^2+n-1}\exp(-\sigma_1(\lambda\wedge\lambda^2)\lambda'), \forall\lambda,\lambda' > 0.
$$

As a result, by combining (33) and Lemma D.1, we get the following bound:

$$
\mathbb{P}(\left\|\hat{\boldsymbol{\alpha}}^{kn+1} - \boldsymbol{\alpha}\right\|_2 + \left\|\hat{B}^{kn+1} - B\right\|_2 > \lambda) \leq C_1(kn)^{n^2+n-1}\exp\left(-\frac{\sigma_0^2\sigma_1\sqrt{kn}}{8n}(\lambda\wedge\lambda^2)\right).
\tag{34}
$$

Now with a similar argument as in the Proof of Theorem 6 in Keskin and Zeevi 2014, we arrive at

$$
\mathbb{E}\left[(\left\|\hat{\boldsymbol{\alpha}}^{kn+1} - \boldsymbol{\alpha}\right\|_2 + \left\|\hat{B}^{kn+1} - B\right\|_2)^2\right] \leq \frac{20C_1 n^5\log(kn+1)}{\sigma_1\sqrt{kn}}.
\tag{35}
$$

In order to use the parameter error to bound the solution error $\Delta_I^t$, we introduce the following Lemma concerning the continuity of constrained strongly convex optimization problems. This lemma establishes Lipschitz continuity of the constrained optimization problem: small errors in estimated parameters $\hat{B}, \hat{\boldsymbol{\alpha}}$ translate to small errors in the optimal solution $\hat{\boldsymbol{d}}$. Combined with the second-order growth condition (Lemma D.3), it allows us to convert parameter estimation bounds into solution quality bounds, which is crucial for analyzing the regret of learning-based algorithms.

**Lemma D.2** (Prop 4.32, Bonnans and Shapiro [2013]). *Suppose the constraint optimization problem*

$$
\begin{aligned}
\max_{p\in\mathcal{P}} \quad & r(\boldsymbol{d}) = \boldsymbol{p}^\top\boldsymbol{d} \\
s.t. \quad & \boldsymbol{d} = \boldsymbol{\alpha} + B\boldsymbol{p}, \\
& A\boldsymbol{d} \leq \frac{\boldsymbol{c}^t}{T-t+1} \\
& \boldsymbol{d} \geq 0,
\end{aligned}
$$

*satisfies second-order growth condition $r(\boldsymbol{d}) \leq r(\boldsymbol{d}^{\pi,t}) - \kappa(\mathsf{Dist}(\boldsymbol{d}, D^{\pi,t}))^2$ for any $d$ in the feasible set and the optimal solution set $D^{\pi,t}$. Then for any optimal solution $\hat{\boldsymbol{d}}$ to the quadratic programming*

$$
\begin{aligned}
\max_{\boldsymbol{p} \in \mathcal{P}} \quad & \boldsymbol{p}^\top \boldsymbol{d} \\
s.t. \quad & \boldsymbol{d} = \hat{\boldsymbol{\alpha}} + \hat{B}\boldsymbol{p}, \\
& Ad \leq \frac{\boldsymbol{c}^t}{T - t + 1} \\
& \boldsymbol{d} \geq 0,
\end{aligned}
$$

*there exists constant $C_2$ such that*

$$
\mathsf{Dist}(\hat{\boldsymbol{d}}, D^{\pi,t}) \leq C_2 \kappa^{-1} \left\| B - \hat{B} \right\|_2
$$

*holds for optimal solution $\hat{\boldsymbol{d}}$ of the second constrained programming problem and all $\hat{B}$ such that $\left\| B - \hat{B} \right\|_2 < \delta$, where $\delta > 0$ depends on $\lambda_{\min}(B + B^\top)$, the minimal eigenvalue of $B + B^\top$.*

Note that, by Lemma D.2, we only need to show that, there exists $\kappa > 0$ such that $r(\boldsymbol{d}) \leq r(\boldsymbol{d}^{\pi,t}) + \kappa(\mathsf{Dist}(\boldsymbol{d}, D^{\pi,t}))^2$ for any $d$ in the feasible set. Fortunately, the following lemma gives the existence of such constant. This lemma verifies the second-order growth condition required by Lemma D.2. It shows that our revenue function $r(\boldsymbol{d})$ is strongly concave around the optimal solution $\boldsymbol{d}^{\pi,t}$, with the growth rate characterized by the minimum eigenvalue $\lambda_{\min}(B + B^\top)$. This strong concavity is what allows parameter estimation errors to translate into bounded solution errors.

**Lemma D.3.** *For any $d$ in the feasible set, we have $r(\boldsymbol{d}) - r(\boldsymbol{d}^{\pi,t}) \geq \frac{1}{4}\lambda_{\min}(B + B^\top)\mathsf{Dist}(\boldsymbol{d}, \boldsymbol{d}^{\pi,t})$.*

By combining Lemma D.2 and D.3, we now have

$$
\left\| \tilde{\boldsymbol{d}}^k - \boldsymbol{d}^{\pi,kn+1} \right\|_2 \leq 4C_2 \lambda_{\min}^{-1}(B + B^\top) \left\| \hat{B}^{kn+1} - B \right\|_2, \tag{36}
$$

By setting $\lambda = \delta$ in (34), we get

$$
\mathbb{P}\left( \left\| \hat{b}^{kn+1} - B \right\|_2 > \delta \right) \leq C_1 (kn)^{n^2+n-1} \exp\left( -\frac{\sigma_0^2 \sigma_1 \sqrt{kn}}{8n}(\delta \wedge \delta^2) \right).
$$

With a similar argument as above and in the proof of Theorem 6 in Keskin and Zeevi 2014, we can get:

$$
\mathbb{E}\left[ \left\| \tilde{\boldsymbol{d}}^k - \boldsymbol{d}^{\pi,kn+1} \right\|_2^2 \right] \leq \frac{C_3 n^5 \lambda_{\min}^{-1}(B + B^\top) \max\{\delta^{-1}, \delta^{-2}\}}{\sigma_1 \sqrt{kn}},
$$

where $C_3$ is some constant determined by $C_1, C_2$. It follows that

$$
\begin{aligned}
\mathbb{E}\left[ \left\| \Delta_I^t \right\|_2^2 \right] &= \mathbb{E}\left[ \left\| \tilde{\boldsymbol{d}}^k - \boldsymbol{d}^{\pi,kn+1} \right\|_2^2 \right] \\
&\leq \frac{C_4 n^5 \lambda_{\min}^{-2}(B + B^\top) \max\{\delta^{-1}, \delta^{-2}\}}{\sigma_1 \sqrt{kn}}
\end{aligned} \tag{37}
$$

for some constant $C_4$. Plugging (31), (32) and (37) into (30) yields:

$$
\begin{aligned}
\mathbb{E}\left[ \left\| \Delta^t \right\|_2^2 \right] &\leq 3 \left( \left\| \Delta_I^t \right\|_2^2 + \left\| \Delta_{II}^t \right\|_2^2 + \left\| \Delta_{III}^t \right\|_2^2 \right) \\
&\leq 3 \left( \frac{4n^2 d_{\max}^2}{(t-1)} + \frac{\sigma_0^2 \left\| B \right\|_2^2}{\sqrt{t}} + \frac{C_3 n^5 \lambda_{\min}^{-1}(B + B^\top) \max\{\delta^{-1}, \delta^{-2}\}}{\sigma_1 \sqrt{kn}} \right) \\
&\leq \frac{C_5}{\sqrt{k}},
\end{aligned} \tag{38}
$$

where $C_5 = 12 \max\{4n^2 d_{\max}^2, \sigma_0^2 \left\| B \right\|_2^2, C_3 \sigma_1^{-1} n^{9/2} \lambda_{\min}^{-1}(B + B^\top) \max\{\delta^{-2}, \delta^{-1}\}\}$.

Now we proceed the argument of (29). With a similar argument of (12), we can get

$$\mathcal{R}^T(\mathbf{Hybrid}^k, \mathcal{F}^T) - \mathcal{R}^T(\mathbf{Hybrid}^{k+1}, \mathcal{F}^T)$$

$$= (T - kn)r(\boldsymbol{d}^{\pi,k}) - \sum_{t=kn+1}^{k+1} r(\boldsymbol{d}^t, \boldsymbol{\epsilon}^t) - (T - (k+1)n)r(\boldsymbol{d}^{\pi,(k+1)n}). \qquad (39)$$

Now we proceed with the following three cases as similar as in Appendix C: Case (I) $\min_i \tilde{d}_i^t \geq \zeta \left[ (T - t + 1)^{-1/4} + t^{-1/4} \right], \forall kn + 1 \leq t \leq (k+1)n$; Case (II): $\max_i \tilde{d}_i^t < \zeta \left[ (T - t + 1)^{-1/4} + t^{-1/4} \right], \forall kn + 1 \leq t \leq (k+1)n, \quad \forall kn + 1 \leq t \leq (k+1)n$; Case (III): others.

**Case (I):** $\min_i \tilde{p}_i^t \geq \zeta \left[ (T - t + 1)^{-1/4} + t^{-1/4} \right], \forall kn + 1 \leq t \leq (k+1)n$. We follow the streamline in Appendix C, except we have different rounding threshold as well as the estimation error. We define $\mathcal{E}_i^t = \{d_i^{\pi,t} \geq \frac{\epsilon_i^t + \Delta_i^t}{T - t}\}$. and $\mathcal{E}^k = \cap_{i=1}^n \cap_{t=kn+1}^{(k+1)n} \mathcal{E}_i^t$.

Recall the decomposition:

$$\mathbb{E}\left[\mathcal{R}^T(\mathbf{Hybrid}^{kn}, \mathcal{F}^T) - \mathcal{R}^T(\mathbf{Hybrid}^{k+1}, \mathcal{F}^T)\right]$$
$$= \mathbb{P}(\mathcal{E}^t)\mathbb{E}\left[\mathcal{R}^T(\mathbf{Hybrid}^t, \mathcal{F}^T) - \mathcal{R}^T(\mathbf{Hybrid}^{t+1}, \mathcal{F}^T)\big|\mathcal{E}^t\right]$$
$$+ \mathbb{P}((\mathcal{E}^t)^c)\mathbb{E}\left[\mathcal{R}^T(\mathbf{Hybrid}^t, \mathcal{F}^T) - \mathcal{R}^T(\mathbf{Hybrid}^{t+1}, \mathcal{F}^T)\big|(\mathcal{E}^t)^c\right] \qquad (40)$$

With a similar argument of deriving (14),we have

$$\mathbb{E}\left[\mathcal{R}^T(\mathbf{Hybrid}^{kn}, \mathcal{F}^T)\mathcal{R}^T(\mathbf{Hybrid}^{(k+1)n}, \mathcal{F}^T)|\mathcal{E}^k\right]$$

$$= \mathbb{E}\left[(T - kn)r(\boldsymbol{d}^{\pi,k}) - \sum_{t=kn+1}^{(k+1)n} r(\boldsymbol{d}^t, \boldsymbol{\epsilon}^t) - (T - (k+1)n)r(\boldsymbol{d}^{\pi,k+1})|\mathcal{E}^k\right]$$

$$\leq \mathbb{E}\left[-\sum_{t=kn+1}^{(k+1)n} (\boldsymbol{\epsilon}^t)^\top \boldsymbol{p}^t - \sum_{t=kn+1}^{(k+1)n} \left[(\Delta^t)^\top B^{-1}\Delta^t + (2\boldsymbol{d}^{\pi,k} - \boldsymbol{\alpha})^\top B^{-1}\Delta^t\right]|\mathcal{E}^k\right]$$

$$+ (T - (k+1)n)\mathbb{E}\left[\frac{(\sum_{t=kn+1}^{(k+1)n} \boldsymbol{\epsilon}^t + \Delta^t)^\top}{T - (k+1)n} B^{-1} \frac{\sum_{t=kn+1}^{(k+1)n} (\boldsymbol{\epsilon}^t + \Delta^t)}{T - (k+1)n} + (2\boldsymbol{d}^{\pi,k} - \boldsymbol{\alpha})^\top B^{-1} \frac{\sum_{t=kn+1}^{(k+1)n} \Delta^t + \boldsymbol{\epsilon}^t}{T - (k+1)n}|\mathcal{E}^k\right]$$

$$= \mathbb{E}\left[(\boldsymbol{d}^{\pi,k})^\top B^{-1} \sum_{t=kn+1}^{(k+1)n} \boldsymbol{\epsilon}^t - \sum_{t=kn+1}^{(k+1)n} (\boldsymbol{\epsilon}^t)^\top B^{-1}\Delta^t|\mathcal{E}^k\right] + \frac{1}{T - (k+1)n}\mathbb{E}\left[(\sum_{t=kn+1}^{(k+1)n} \boldsymbol{\epsilon}^t + \Delta^t)^\top B^{-1}(\sum_{t=kn+1}^{(k+1)n} \boldsymbol{\epsilon}^t + \Delta^t)|\mathcal{E}^k\right]$$

$$(41)$$

Note that $\boldsymbol{\epsilon}^t$ is independent of $\Delta^t$. We have $\mathbb{E}\left[\boldsymbol{\epsilon}^t B^{-1}\Delta^t\right] = 0$. Note that $\Delta^t \leq \boldsymbol{d}_{\max}$. Following the proof of (15) and (16) leads to

$$\mathbb{P}(\mathcal{E}^k)\mathbb{E}\left[(\boldsymbol{d}^{\pi,k})^\top B^{-1} \sum_{t=kn+1}^{(k+1)n} \boldsymbol{\epsilon}^t - \sum_{t=kn+1}^{(k+1)n} (\boldsymbol{\epsilon}^t)^\top B^{-1}\Delta^t|\mathcal{E}^k\right]$$

$$\leq 2\left\|B^{-1}\right\|_2 d_{\max}\mathbb{P}((\mathcal{E}^k)^c)^{1/2}\mathbb{E}\left[\sum_{t=kn+1}^{(k+1)n} \left\|\boldsymbol{\epsilon}^t\right\|_2^2\right]^{1/2}$$

$$\leq 2\sigma \left\|B^{-1}\right\|_2 d_{\max}\mathbb{P}((\mathcal{E}^k)^c)^{1/2} \qquad (42)$$

and

$$\mathbb{P}(\mathcal{E}^k)\mathbb{E}\left[(\sum_{t=kn+1}^{(k+1)n}\boldsymbol{\epsilon}^t+\Delta^t)^\top B^{-1}(\sum_{t=kn+1}^{(k+1)n}\boldsymbol{\epsilon}^t+\Delta^t)|\mathcal{E}^k\right]$$

$$\leq \frac{2n\left\|B^{-1}\right\|_2}{T-(k+1)n}\sum_{t=kn+1}^{(k+1)n}\mathbb{E}\left[\left\|\boldsymbol{\epsilon}^t\right\|_2^2+\left\|\Delta^t\right\|_2^2\right]$$

$$\overset{(a)}{\leq}\frac{2n\left\|B^{-1}\right\|_2}{T-(k+1)n}\sum_{t=kn+1}^{(k+1)n}\left(\sigma^2+C_5 k^{-1/2}\right)$$

$$=\frac{2n^2\left\|B^{-1}\right\|_2}{T-(k+1)n}\left(\sigma^2+C_5 k^{-1/2}\right),\tag{43}$$

where we apply (38) in (a).

Now following the deduction in (17), we get

$$\mathbb{P}((\mathcal{E}^t)^c)\mathbb{E}\left[\mathcal{R}^T(\mathbf{Hybrid}^t,\mathcal{F}^T)-\mathcal{R}^T(\mathbf{Hybrid}^{t+1},\mathcal{F}^T)\big|(\mathcal{E}^t)^c\right]$$

$$\leq (T-kn)r_{\max}\mathbb{P}((\mathcal{E}^k)^c)+\sqrt{n}\sigma\mathbb{P}((\mathcal{E}^k)^c)^{1/2}.\tag{44}$$

Now combining (40), (41), (42), (43) and (44), we get

$$\mathbb{E}\left[\mathcal{R}^T(\mathbf{Hybrid}^t,\mathcal{F}^T)-\mathcal{R}^T(\mathbf{Hybrid}^{t+1},\mathcal{F}^T)\right]$$

$$2\sigma\left\|B^{-1}\right\|_2 d_{\max}\mathbb{P}((\mathcal{E}^k)^c)^{1/2}+\frac{2n^2\left\|B^{-1}\right\|_2}{T-(k+1)n}(\sigma^2+C_5 k^{-1/2})+(T-kn)r_{\max}\mathbb{P}((\mathcal{E}^k)^c)+\sqrt{n}\sigma\mathbb{P}((\mathcal{E}^k)^c)^{1/2}.$$
$$\tag{45}$$

Now we give bound to $\mathbb{P}((\mathcal{E}^t)^c)$. We consider two events: $\mathcal{I}_{i1}^t=\{\zeta((T-t+1)^{-1/4}+t^{-1/4})/3\geq |\Delta_i^t|\}$; $\mathcal{I}_{i2}^t=\{\zeta((T-t+1)^{-1/4}+t^{-1/4})/3\geq\frac{\epsilon_i^t}{T-(k+1)n}\}$. Then we have

$$\mathbb{P}((\mathcal{E}^k)^c)\leq 1-\mathbb{P}(\cap_{i=1}^n\cap_{t=kn+1}^{(k+1)n}(\mathcal{I}_{i1}^t\cap\mathcal{I}_{i2}^t))$$

$$\leq\sum_{i=1}^n\sum_{t=kn+1}^{(k+1)n}(\mathbb{P}(\mathcal{I}_{i1}^t)+\mathbb{P}(\mathcal{I}_{i2}^t)).$$

With (36), (31), (32) and (34), we have

$$\mathbb{P}((\mathcal{I}_{i1}^t)^c)$$

$$\leq\mathbb{P}\left(4C_2\lambda_{\min}^{-1}(B+B^\top)\left\|\hat{B}^{kn+1}-B\right\|_2+\frac{2n}{t-1}d_{\max}+\sigma_0\left\|B\right\|_2 t^{-1/4}>\zeta((T-t+1)^{-1/4}+t^{-1/4})/3\right)$$

$$\leq\mathbb{P}\left(C_2\lambda_{\min}^{-1}(B+B^\top)\left\|\hat{B}^{kn+1}-B\right\|_2>\zeta((T-t+1)^{-1/4}+t^{-1/4})/24\right)$$

$$\leq C_1(kn)^{n^2+n-1}\exp\left(-\frac{\sigma_0^2\sigma_1\sqrt{kn}}{8n}(\lambda\wedge\lambda^2)\right)$$

$$\leq C_5^2/(n^2 T^2)\tag{46}$$

for some constant $C_5$, where $\lambda=C_2^{-1}\lambda_{\min}(B+B^\top)\zeta((T-t+1)^{-1/4}+t^{-1/4})/24$ and we have

$$\lambda^2\wedge\lambda\geq\frac{8(n^{5/2}+n)\log^3 n\log^2 T}{k^{-1/2}\sigma_0^2\sigma}$$

by definition of $\zeta$. For the event $\mathcal{I}_{i2}^c$, it follows directly from the derivation of (19) that

$$\mathbb{P}((\mathcal{I}_{i2}^t)^c)$$

$$\leq\mathbb{P}\left(\zeta((T-t+1)^{-1/4}+t^{-1/4})/3<\frac{\epsilon_i^t}{T-(k+1)n}\right)$$

$$\leq\exp\left(-\frac{\zeta^2(T-(k+1)n)^2}{36\sigma^2}\right)$$

$$\leq C_5^2/(n^2 T^2).\tag{47}$$

Now combining (46) and (47), we get

$$\mathbb{P}((\mathcal{E}^k)^c) \leq \sum_{i=1}^{n} \sum_{t=kn+1}^{(k+1)n} (\mathbb{P}(\mathcal{I}_{i1}^t) + \mathbb{P}(\mathcal{I}_{i2}^t))$$
$$\leq 2C_5^2/T^2.$$

Plugging the above inequality into (45) yields

$$\mathbb{E}\left[\mathcal{R}^T(\mathbf{Hybrid}^k, \mathcal{F}^T) - \mathcal{R}^T(\mathbf{Hybrid}^{k+1}, \mathcal{F}^T)\right]$$

$$\leq 2C_5\sigma \left\|B^{-1}\right\|_2 d_{\max} T^{-1} + \frac{2n^2 \left\|B^{-1}\right\|_2}{T - (k+1)n}(\sigma^2 + C_5 k^{-1/2}) + C_5^2(T - kn)r_{\max}/T^2 + C_5\sqrt{n}\sigma T^{-1}. \tag{48}$$

**Case (II):** $\max_i \tilde{d}_i^t < \zeta\left[(T - t + 1)^{-1/4} + t^{-1/4}\right], \forall kn + 1 \leq t \leq (k+1)n, \quad \forall kn + 1 \leq t \leq (k+1)n$. This case is much simpler, just follow (21):

$$\mathbb{E}\left[\mathcal{R}^T(\mathbf{Hybrid}^k, \mathcal{F}^T) - \mathcal{R}^T(\mathbf{Hybrid}^{k+1}, \mathcal{F}^T)\right]$$

$$\leq \mathbb{E}\left[(T - kn)r(\boldsymbol{d}^{\pi,k}) - (T - (k+1)n)r\left(\frac{T - kn}{T - (k+1)n}\boldsymbol{d}^{\pi,k}\right)\right]$$

$$= \mathbb{E}\left[(T - kn)(\boldsymbol{d}^{\pi,k})^\top B^{-1}(\boldsymbol{d}^{\pi,t} - \boldsymbol{\alpha}) - (T - (k+1)n)\frac{T - kn}{T - (k+1)n}(\boldsymbol{d}^{\pi,t})^\top B^{-1}(\frac{T - kn}{T - (k+1)n}\boldsymbol{d}^{\pi,t} - \boldsymbol{\alpha})\right]$$

$$= \mathbb{E}\left[-\frac{T - kn}{T - (k+1)n}(\boldsymbol{d}^{\pi,t})^\top B^{-1}\boldsymbol{d}^{\pi,t}\right]$$

$$= -\frac{T - kn}{T - (k+1)n}\mathbb{E}\left[(\boldsymbol{d}^{\pi,t} + \Delta^t)^\top B^{-1}(\boldsymbol{d}^t + \Delta^)\right]$$

$$\leq -\frac{2(T - kn)}{T - (k+1)n}\mathbb{E}\left[(\boldsymbol{d}^{kn+1})^\top B^{-1}\boldsymbol{d}^{kn+1}\right] - \frac{2(T - kn)}{T - (k+1)n}\mathbb{E}\left[(\Delta^{kn+1})^\top B^{-1}\Delta^{kn+1}\right]$$

$$\leq 8\zeta^2\left[(T - kn + 1)^{-1/2} + (kn)^{-1/2}\right] + \frac{8C_5 \left\|B^{-1}\right\|_2}{\sqrt{k}}. \tag{49}$$

where we again use (38) in the last line.

**Case (III): others.** Following the same streamline of the argument in Appendix C and utilizing (48), (49), we can get

$$\mathbb{E}\left[\mathcal{R}^T(\mathbf{Hybrid}^k, \mathcal{F}^T) - \mathcal{R}^T(\mathbf{Hybrid}^{k+1}, \mathcal{F}^T)\right]$$

$$\leq 8\zeta^2\left[(T - kn + 1)^{-1/2} + (kn)^{-1/2}\right] + \frac{8C_5 \left\|B^{-1}\right\|_2}{\sqrt{k}}$$

$$+ 2C_5\sigma \left\|B^{-1}\right\|_2 d_{\max} T^{-1} + \frac{2n^2 \left\|B^{-1}\right\|_2}{T - (k+1)n}(\sigma^2 + C_5 k^{-1/2}) + C_5^2(T - kn)r_{\max}/T^2 + C_5\sqrt{n}\sigma T^{-1}. \tag{50}$$

**Wrap-up.** Combinine (48), (49) and (50) leads to

$$\mathbb{E}\left[\mathcal{R}^T(\mathbf{Hybrid}^k, \mathcal{F}^T) - \mathcal{R}^T(\mathbf{Hybrid}^{k+1}, \mathcal{F}^T)\right]$$

$$\leq 8\zeta^2\left[(T - kn + 1)^{-1/2} + (kn)^{-1/2}\right] + \frac{8C_5 \left\|B^{-1}\right\|_2}{\sqrt{k}}$$

$$+ 2C_5\sigma \left\|B^{-1}\right\|_2 d_{\max} T^{-1} + \frac{2n^2 \left\|B^{-1}\right\|_2}{T - (k+1)n}(\sigma^2 + C_5 k^{-1/2}) + C_5^2(T - kn)r_{\max}/T^2 + C_5\sqrt{n}\sigma T^{-1}.$$

Plugging this into (40) yields

$$\text{Regret}^T(\pi) = \sum_{k=1}^{T'} \mathbb{E}\left[\mathcal{R}^T(\textbf{Hybrid}^k, \mathcal{F}^T) - \mathcal{R}^T(\textbf{Hybrid}^{k+1}, \mathcal{F}^T)\right]$$
$$= O\big(\big[(\zeta^2 + C_5 \left\| B^{-1}\right\|_2\big] \sqrt{T}\big).$$

## D.1 Proof of Proposition 4.1

**Proof roadmap** We establish the lower bound $\Omega(\max\{\rho\sqrt{T}, (\epsilon^0)^2 T\})$ by constructing two adversarial problem instances:

1. **Instance Construction:** Define two parameter sets $(\boldsymbol{\alpha}, B)$ and $(\boldsymbol{\alpha}', B')$ that are:
   - Statistically indistinguishable when $(\epsilon^0)^2 T \ll \sqrt{T}$
   - Require fundamentally different pricing strategies
   - Force any algorithm to suffer large regret on at least one instance

2. **Information-Theoretic Argument:** Use KL divergence bounds to show that any policy cannot distinguish between the two instances with high confidence when data is limited, establishing the $\Omega((\epsilon^0)^2 T)$ term.

3. **Reduction to No-Information Lower Bound:** For the $\Omega(\rho\sqrt{T})$ term, reduce to the known lower bound from Lemma 3.3 by choosing $\epsilon^0 = \Theta(T^{-(1-\gamma)/2})$ to match the minimax rate.

**Implication:** This result directly follows from Proposition 4.1 by appropriate parameter scaling. It confirms that Theorem 4.2 achieves the optimal rate, and no algorithm can improve upon it without additional assumptions.

The proof is brief and follows immediately from prior results.

We follow Sec 15.2 Lattimore and Szepesvári [2020] and Cheung and Lyu [2024]. Consider two demand functions

$$\boldsymbol{d}_\theta(\boldsymbol{p}) = 2\Delta - \Delta\boldsymbol{p} + \boldsymbol{\epsilon}, \quad \boldsymbol{d}_{\theta'}(\boldsymbol{p}) = 3\Delta - 2\Delta\boldsymbol{p} + \boldsymbol{\epsilon},$$

with $\Delta = T^{-\beta}$, where $\boldsymbol{\epsilon} \sim \mathcal{N}(0, 1)$.

Assume that both $\theta, \theta'$ have the same offline data distribution $\mathbb{P}^{\textbf{off}}$. Denote $P_\pi^{\textbf{on}}, Q_\pi^{\textbf{on}}$ as the online distribution of the demands under policy $\pi$. Additionally, define $P, Q$ as the joint distribution of $(P^{\textbf{off}}, P_\pi^{\textbf{on}})$ and $(P^{\textbf{off}}, Q_\pi^{\textbf{on}})$, respectively.

Then

$$\text{Regret}(\pi)\theta = \mathbb{E}_P\left[\sum_{t=1}^T \Delta^{-1}(\Delta\boldsymbol{p}^t - \Delta)^2\right], \quad \text{Regret}(\pi)\theta' = \mathbb{E}_Q\left[\sum_{t=1}^T 2\Delta^{-1}\big(\Delta\boldsymbol{p}^t - \tfrac{3\Delta}{2}\big)^2\right].$$

For set $S \subset \mathbb{R}$, denote $N_T(S)$ as the total number of $p_t$ such that $p_t \in S$. Consider the event

$$\mathcal{I} = \left\{ N_T\big([\tfrac{5\Delta}{4}, \infty)\big) > \tfrac{T}{2} \right\}.$$

Then by Bretagnolle–Huber inequality Lattimore and Szepesvári, 2020, Thm 14.2 we have

$$\text{Regret}(\pi, \theta) + \text{Regret}(\pi, \theta')$$
$$\geq \frac{\Delta}{16} * \frac{T}{2} * P(\mathcal{I}) + \frac{2\Delta}{16} * \frac{T}{2} * Q(\mathcal{I}^C)$$
$$\geq \frac{T^{1-\gamma}}{32}(P(\mathcal{I}) + Q(\mathcal{I}^C))$$
$$\geq \frac{T^{1-\gamma}}{32} \exp(-\text{KL}(P \| Q)).$$

For Gaussian $P_\theta(\boldsymbol{p}) = \mathcal{N}(2\Delta - \Delta\boldsymbol{p}, 1)$ and $Q_{\theta'}(\boldsymbol{p}) = \mathcal{N}(3\Delta - 2\Delta\boldsymbol{p}, 1)$, we have $\mathsf{KL}\big(P_\theta \,\|\, P_{\theta'}\big) = \frac{(\Delta - p)^2}{2}$. In order to to further give bound to the term $\exp(-\mathsf{KL}\big(P \,\|\, Q\big))$, we introduce the following lemma in Cheung and Lyu [2024]. This lemma shows that when two problem instances share the same offline data distribution but differ in their online dynamics, the KL divergence between the full joint distributions (offline + online) equals the KL divergence of only the online parts. This factorization is crucial for constructing adversarial instances: it allows us to design two indistinguishable problems based on offline data that require fundamentally different online strategies, establishing information-theoretic lower bounds.

**Lemma D.4.** *Consider two instances with onlien distribution $P$, $Q$ and shared offline dataset with samples $\{(p^{-t}, \boldsymbol{d}^{-t})\}_{t=1}^N$, then for any admissible policy $\pi$, it holds that*

$$\exp(-\mathsf{KL}\big(P \,\|\, Q\big)) = \exp(-\mathbb{E}_P\left[\sum_{t=1}^T \mathsf{KL}\big(P_\theta(p^t) \,\|\, Q_{\theta'}(p^t)\big)\right].$$

By Lemma D.4, we have

$$\begin{aligned}
\exp(-\mathsf{KL}\big(P \,\|\, Q\big)) &= \exp(-\mathbb{E}_P\left[\sum_{t=1}^T \mathsf{KL}\big(P_\theta(p^t) \,\|\, Q_{\theta'}(p^t)\big)\right] \\
&= \exp(-\mathbb{E}_P\left[\sum_{t=1}^T \frac{(\Delta - \Delta\boldsymbol{p}^t)^2}{2}\right] \\
&= \exp(-\Delta\mathsf{Regret}\big(\pi, \theta\big)) \\
&\geq \exp(-\Delta C T^\gamma) \\
&= \exp(-C).
\end{aligned}$$

As a result, we have

$$\mathsf{Regret}\big(\pi, \theta\big) + \mathsf{Regret}\big(\pi, \theta'\big) \geq \frac{T^{1-\gamma}}{32}\exp(-C) = \Omega(T^{1-\gamma}).$$

Since $\beta \in [0, \frac{1}{2})$, we know that $\mathsf{Regret}\big(\pi, \theta'\big) \geq \Omega(T^{1-\gamma})$.

# E    Proof of Theorem 4.2

**Proof Roadmap**    We prove the regret bound $O(\min\{\rho\sqrt{T}, (\epsilon^0)^2 T + C'\log T\})$ by leveraging the informed price-demand pair $(p^0, d^0)$:

1. **Estimate-Then-Select Decision:** Algorithm 3 first decides whether to use the informed price based on comparing $(\epsilon^0)^2 T$ versus $\rho\sqrt{T}$:

   - If $(\epsilon^0)^2 T > \rho\sqrt{T}$: Switch to Algorithm 2 (no-information setting)
   - If $(\epsilon^0)^2 T \leq \rho\sqrt{T}$: Exploit the informed price with anchored regression

   We focus on the second case, as the first case reduces to Theorem 3.1.

2. **Anchored Regression Framework:** Define the improved data variance quantity:

$$J^t := n^{-1}\sum_{s=1}^t \|p^s - p^0\|_2^2 \geq tn^{-1}\delta_0^2$$

   where $\delta_0 = \|p^\star - p^0\|_2$ measures distance from informed price to optimal price. This replaces the $O(t^{-1/2})$ variance growth in Appendix D with $O(t)$ *linear* growth, dramatically improving estimation accuracy.

3. **Enhanced Parameter Estimation Bound:** Using the anchored regression:

$$\min_{\boldsymbol{\alpha}, B}\sum_t \|\boldsymbol{d}^t - (\boldsymbol{\alpha} + B\boldsymbol{p}^t)\|^2 \quad \text{subject to} \quad \|\boldsymbol{d}^0 - (\boldsymbol{\alpha} + Bp^0)\|^2 \leq (\epsilon^0)^2$$

   we obtain improved bounds:

- $\mathbb{E}[\|\hat{\boldsymbol{\alpha}}^t - \boldsymbol{\alpha}\|_2^2 + \|\hat{B}^t - B\|_2^2] = O(\log t/\sqrt{t})$ (vs. $O(n^5/\sqrt{k})$ without informed prior)
- Misspecification contributes additive $(\epsilon^0)^2$ term per period

4. **Modified Single-Step Analysis:** Follow the three-case framework from Appendix C, but with:

   - Rounding threshold: $\zeta[(T-t+1)^{-1/2} + t^{-1/2}]$ (tighter than no-info case)
   - Error decomposition: $\Delta^t = \Delta_I^t + \Delta_{II}^t$ (no exploration noise $\Delta_{III}^t$ due to anchoring)
   - Improved $\Delta_I^t$ bound from step 3

5. **Final Aggregation:** Sum over all $T$ periods:

   - Estimation error: $\sum_{t=1}^T O(\log t/\sqrt{t}) = O(\log T)$ (logarithmic due to linear variance growth!)
   - Misspecification: $\sum_{t=1}^T 2(\epsilon^0)^2 = O((\epsilon^0)^2 T)$
   - Rounding and noise: $O(\zeta^2 \log T + \sigma^2 \|B^{-1}\|_2 \log T)$

   Combining yields $O((\epsilon^0)^2 T + C' \log T)$ where $C' = \sigma_0 d_{\max}\|B^{-1}\|_2 + n^2\sigma^2\|B^{-1}\|_2 + \zeta^2$.

**Key Insight:** The informed price $(p^0, d^0)$ acts as an *anchor* that accelerates parameter learning from $\tilde{O}(\sqrt{T})$ to $O(\log T)$ when $\epsilon^0$ is small. The trade-off is explicit: accurate predictions ($\epsilon^0 \to 0$) yield near-optimal $O(\log T)$ regret, while poor predictions ($\epsilon^0$ large) gracefully degrade to $O(\sqrt{T})$ regret.

**Comparison to Appendix D:** The main technical difference is the variance analysis: linear growth $J^t \geq tn^{-1}\delta_0^2$ (enabled by the informed price) versus sublinear growth $J^t = O(\sqrt{t})$ (from exploration noise alone). This distinction propagates through the entire proof.

The detailed proof follows below.

In this section, we follow a similar streamline as in Appendix D. We only need to consider the case $(\epsilon^0)^2 T \leq \rho\sqrt{T}$ and the informed prices are utilized. To begin with, recall our re-solve constrained programming problem:

$$
\begin{aligned}
\max_{p \in \mathcal{P}} \quad & r = \boldsymbol{p}^\top \boldsymbol{d} \\
\text{s.t.} \quad & \boldsymbol{d} = \boldsymbol{\alpha} + B\boldsymbol{p}, \\
& A\boldsymbol{d} \leq \frac{\boldsymbol{c}^t}{T-t+1},
\end{aligned}
\tag{51}
$$

where $\boldsymbol{c}^t$ is the inventory level at the beginning of time $t$. We use the single-step difference decomposition:

$$
\text{Regret}^T(\pi) = \mathbb{E}\left[\sum_{t=1}^T \mathcal{R}^T(\mathbf{Hybrid}^t, \mathcal{F}^T) - \mathcal{R}^T(\mathbf{Hybrid}^{t+1}, \mathcal{F}^T)\right]
$$

$$
= \sum_{t=1}^T \mathbb{E}\left[\mathcal{R}^T(\mathbf{Hybrid}^t, \mathcal{F}^T) - \mathcal{R}^T(\mathbf{Hybrid}^{t+1}, \mathcal{F}^T)\right].
\tag{52}
$$

We proceed by giving bound to the term $\mathbb{E}\left[\mathcal{R}^T(\mathbf{Hybrid}^t, \mathcal{F}^T) - \mathcal{R}^T(\mathbf{Hybrid}^{t+1}, \mathcal{F}^T)\right]$ for $1 \leq t \leq T$. We define $\Delta^t = \boldsymbol{d}^t - \boldsymbol{d}^{\pi,t}$ similarly. Let $l = \mod(t, n)$. Then we have

$$
\Delta^t = \underbrace{(\tilde{\boldsymbol{d}}^t - \boldsymbol{d}^t)}_{:=\Delta_I^t} + \underbrace{\sigma_0 t^{-1/2} B e_l}_{:=\Delta_{II}^t}.
\tag{53}
$$

It follows directly that

$$
\left\|\Delta_{II}^t\right\|_2 \leq \sigma_0 t^{-1/2}\|B\|_2.
\tag{54}
$$

In order to give bound to the term $\Delta_I^t$, we define the quanity $J^t$ as

$$J^t := n^{-1} \sum_{s=1}^t \left\| p^s - p^0 \right\|_2^2 \geq^{-1} \sum_{s=1}^t \left\| p^\star - p^0 \right\|_2^2 = t n^{-1} \delta_0^2. \tag{55}$$

Now we introduce the following tail bound on the parameter estimation analogous to Lemma D.1. Similar to Lemma D.1 but for the anchored regression setting with informed price $(p^0, d^0)$. The key difference is that the data variance $J^t$ now grows *linearly* in $t$ (rather than $\sqrt{t}$) because the informed price serves as an anchor point, dramatically improving estimation accuracy. This is the technical reason why the informed-price algorithm achieves $O(\log T)$ regret when $\epsilon^0$ is small.

**Lemma E.1** (Keskin and Zeevi 2014). *Under our policy and estimation* (**??**), *if $\epsilon^0 = 0$ (i.e. no misspecification), then there exist finite positive constants $C_6, \sigma_2$ such that*

$$\mathbb{P}(\left\| \hat{B}^t - B \right\|_2 > \lambda, J^t \geq \lambda') \leq C_6 t^{n^2-1} \exp(-\sigma_2(\lambda \wedge \lambda^2) J^t).$$

Moreover, we have the following lower bound on the minimal eigenvalue of the matrix $\hat{P}^t := \sum_{s=1}^{t-1}(p^s - p^0)(p^s - p^0)^\top$ in (**??**). This lemma provides a lower bound on the minimal eigenvalue of the empirical covariance matrix $\hat{P}^t = \sum_{s=1}^{t-1}(p^s - p^0)(p^s - p^0)^\top$ in terms of the variance quantity $J^t$. This ensures that the regression problem is well-conditioned: sufficient price exploration (measured by $J^t$) guarantees that the Fisher information matrix is invertible, enabling accurate parameter estimation.

**Lemma E.2** (Keskin and Zeevi 2014). *Under our policy, we have $\lambda_{\min}(\hat{P}^t) \geq J^t$.*

Note that $\hat{P}^t$ is positive definite by the deduction above. Combining (**??**), Lemma E.1 and E.2 leads to

$$\mathbb{P}(\left\| \hat{B}^t - B \right\|_2 > \lambda + \epsilon^0 \leq C_7 t^{n^2-1} \exp(-\sigma_2(\lambda \wedge \lambda^2) J^t) \tag{56}$$

for some constant $C_7$ under offline data misspecification $\epsilon^0$. Now following a similar argument as in the proof of (37) (except that we have the $J^t$ linear in $T$, rather than square root order of $T$ in the previous section, hence we have better bound here), we can get

$$\mathbb{E}\left[ \left\| \Delta_I^t \right\|_2^2 \right] \leq \frac{C_8 n^5 \lambda_{\min}^{-1}(B + B^\top) \max\{\delta^{-1}, \delta^{-2}\}}{\sigma_2 t} + (\epsilon^0)^2. \tag{57}$$

Plugging (54) and (57) into (53)

$$
\begin{aligned}
\mathbb{E}\left[ \left\| \Delta^t \right\|_2^2 \right] &\leq 2 \left( \left\| \Delta_I^t \right\|_2^2 + \left\| \Delta_{II} \right\|_2^2 \right) \\
&\leq 2 \left( \frac{C_8 n^5 \lambda_{\min}^{-1}(B + B^\top) \max\{\delta^{-1}, \delta^{-2}\}}{\sigma_2 t} + (\epsilon^0)^2 + \sigma_0^2 t^{-1} \left\| B \right\|_2^2 \right) \\
&\leq \frac{C_9}{t}, \tag{58}
\end{aligned}
$$

where $C_9 = 8 \max\{C_8 n^5 \lambda_{\min}^{-1}(B + B^\top) \max\{\delta^{-1}, \delta^{-2}\}/\sigma_2, \sigma_0^2 \left\| B \right\|_2^2\}$.

Now we proceed with almost the same argument as in Appendix D. We consider three cases: Case (I) $\min_i \tilde{d}_i^t \geq \zeta \left[ (T - t + 1)^{-1/2} + t^{-1/2} \right]$; Case (II) $\max_i \tilde{d}_i^t \leq \zeta \left[ (T - t + 1)^{-1/2} + t^{-1/2} \right]$; Case (III) others.

**Case (I).** We follow a almost same argument as the derivation of (48) except by replacing the bound of $\mathbb{E}\left[ \left\| \Delta \right\|_2^2 \right]$ by (58) and the tail bound of (46), (47) with corresponding rounding threshold $\zeta \left[ (T - t + 1)^{-1/2} + t^{-1/2} \right]$ and $J^t$ in (55). We can get

$$\mathbb{E}\left[ \mathcal{R}^T(\mathbf{Hybrid}^t, \mathcal{F}^T) - \mathcal{R}^T(\mathbf{Hybrid}^{t+1}, \mathcal{F}^T) \right]$$

$$4 C_9 \sigma_0 \left\| B^{-1} \right\| \boldsymbol{d}_{\max} T^{-1} + \frac{4 n^2 \left\| B^{-1} \right\|_2}{T - t} (\sigma^2 + C_9 T^{-1}) + 2 C_9^2 (T - t + 1) r_{\max} / T^2 + C_0 \sqrt{n} \sigma T^{-1} + 2(\epsilon^0)^2. \tag{59}$$

**Case (II).** With a similar argument of deriving (49) by replacing the rounding threshold with $\zeta\left[(T-t+1)^{-1/2}+t^{-1/2}\right]$, we can get

$$\mathbb{E}\left[\mathcal{R}^T(\mathbf{Hybrid}^t,\mathcal{F}^T)-\mathcal{R}^T(\mathbf{Hybrid}^{t+1},\mathcal{F}^T)\right]$$
$$\leq 8\zeta^2\left[(T-t+1)^{-1}+t^{-1}\right]+\frac{8C_9\left\|B^{-1}\right\|_2}{t}. \tag{60}$$

**Case (III).** With a similar argument of deriving (61), we arrive at

$$\mathbb{E}\left[\mathcal{R}^T(\mathbf{Hybrid}^t,\mathcal{F}^T)-\mathcal{R}^T(\mathbf{Hybrid}^{t+1},\mathcal{F}^T)\right]$$
$$\leq 4C_9\sigma_0\left\|B^{-1}\right\|\boldsymbol{d}_{\max}T^{-1}+\frac{4n^2\left\|B^{-1}\right\|_2}{T-t}(\sigma^2+C_9T^{-1})+2C_9^2(T-t+1)r_{\max}/T^2+C_0\sqrt{n}\sigma T^{-1}+2(\boldsymbol{\epsilon}^0)^2$$
$$+8\zeta^2\left[(T-t+1)^{-1}+t^{-1}\right]+\frac{8C_9\left\|B^{-1}\right\|_2}{t}. \tag{61}$$

**Wrap-up.** Now combining (59), (60) and (61) yields

$$\mathbb{E}\left[\mathcal{R}^T(\mathbf{Hybrid}^t,\mathcal{F}^T)-\mathcal{R}^T(\mathbf{Hybrid}^{t+1},\mathcal{F}^T)\right]$$
$$\leq 4C_9\sigma_0\left\|B^{-1}\right\|\boldsymbol{d}_{\max}T^{-1}+\frac{4n^2\left\|B^{-1}\right\|_2}{T-t}(\sigma^2+C_9T^{-1})+2C_9^2(T-t+1)r_{\max}/T^2+C_0\sqrt{n}\sigma T^{-1}+2(\boldsymbol{\epsilon}^0)^2$$
$$+8\zeta^2\left[(T-t+1)^{-1}+t^{-1}\right]+\frac{8C_9\left\|B^{-1}\right\|_2}{t}.$$

Now plugging the above inequality into (52) leads to

$$\mathsf{Regret}^T(\pi)=\sum_{t=1}^T\mathbb{E}\left[\mathcal{R}^T(\mathbf{Hybrid}^t,\mathcal{F}^T)-\mathcal{R}^T(\mathbf{Hybrid}^{t+1},\mathcal{F}^T)\right]$$
$$=O\left(C_{10}\log T+(\boldsymbol{\epsilon}^0)^2T\right),$$

where $C_{10}=2C_0\sigma_0\left\|B^{-1}\right\|_2 \boldsymbol{d}_{\max}+2n^2\left\|B^{-1}\right\|_2\sigma^2+C_0\sqrt{n}\sigma+16\zeta^2+8C_9\left\|B^{-1}\right\|_2$.

## F  Proof of Proposition 4.3

This is a direct result following Proposition 4.1. We just need to take $\boldsymbol{\epsilon}$ of order $T^{-(1-\gamma)/2}$.

## G  Proof of lemmas

### G.1  Proof of Lemma D.3

We rewrite the target as $r(\boldsymbol{d})=\boldsymbol{d}^\top B^{-1}(\boldsymbol{d}-\boldsymbol{\alpha}$, which is a quadratic function since $B$ is positive definite. For optimal solution $\boldsymbol{d}^{\pi,t}$, $\boldsymbol{d}-\boldsymbol{d}^{\pi,t}$ is an descending direction for feasible $d$ and it follows from Boyd and Vandenberghe [2004] that $(\boldsymbol{d}^\top-\boldsymbol{d}^{\pi,t})\nabla_{\boldsymbol{d}^{\pi,t}}r(\boldsymbol{d}^{\pi,t})\leq 0$. Moreover, with direct calculation, we can get

$$r(\boldsymbol{d})=r(\boldsymbol{d}^{\pi,t})+(\boldsymbol{d}-\boldsymbol{d}^{\pi,t})^\top\nabla_{\boldsymbol{d}^{\pi,t}}r(\boldsymbol{d}^{\pi,t})+\frac{1}{2}(\boldsymbol{d}-\boldsymbol{d}^{\pi,t})^\top B^{-1}(\boldsymbol{d}-\boldsymbol{d}^{\pi,t}).$$

As a result, we have

$$r(\boldsymbol{d})\leq r(\boldsymbol{d}^{\pi,t})+\frac{1}{2}(\boldsymbol{d}-\boldsymbol{d}^{\pi,t})^\top B^{-1}(\boldsymbol{d}-\boldsymbol{d}^{\pi,t})$$
$$\leq r(\boldsymbol{d}^{\pi,t})+\frac{1}{4}\lambda_{\min}(B^{-1}+B^{-\top})\left\|\boldsymbol{d}-\boldsymbol{d}^{\pi,t}\right\|_2^2$$
$$\leq r(\boldsymbol{d}^{\pi,t})+\frac{1}{4}\lambda_{\min}(B^{-1}+B^{-\top}).$$

