# OpenReview forum: "Learning to price with resource constraints: from full information to machine-learned prices"
_NeurIPS.cc/2025/Conference — NeurIPS 2025 poster_

### Official Review · Reviewer_8Fer · 2025-06-23

**Clarity:** 3
**Significance:** 3
**Originality:** 3
**Rating:** 4
**Confidence:** 3

**Summary:**

This paper addresses dynamic pricing with resource constraints by developing three algorithms tailored to different levels of prior information about the demand function. For the full-information setting, where the demand function is known, the authors propose a Boundary Attracted Re-solve Method that achieves O(log T) regret without requiring conditions found in prior work. In the no-information setting, an online learning algorithm is presented that attains optimal regret by balancing exploration and exploitation. For the intermediate informed-price setting, where a machine-learned price estimate with a known error bound (ϵ_0) is available, the paper introduces an estimate-then-select algorithm. This algorithm achieves a regret of O(min{sqrt(T),(ϵ_0)2T}+log T), effectively bridging the performance gap between the full and no-information scenarios. The paper supports these theoretical findings with numerical experiments.

**Questions:**

For NeurIPS, I think it would be valuable to give a little bit of practical context about the dynamic pricing problem studied here — i.e., what are some applications that this problem models (or approximately models)?

**Ethical Concerns:**

["NO or VERY MINOR ethics concerns only"]

**Final Justification:**

In the author discussion phase, my concerns relating to the numerical study and practical context were addressed well. After reading other reviews and responses, I decided to maintain my score -- I hope to see revisions that address my comments and other reviewers' comments in the final version of the paper.

**Limitations:**

Yes.

**Quality:**

3

**Strengths And Weaknesses:**

**Strengths**
The paper overall presents sound and novel results relating to the studied dynamic pricing problem, and is well-written.
I appreciate the examination of improvements that are attainable when the seller has access to an initial machine-learned estimate of the price-demand, including the impossibility result when the error bound $\epsilon_0$ is unknown.

**Weaknesses**
As far as I can tell, the numerical study does not compare against any baselines from prior work, and the instances are relatively small and randomly generated, which limits their appeal somewhat.  Moreover, the figures in the experiment section are quite hard to read — axes, legends, and labels should all be resized for this format.

---

> ### Author Rebuttal · Authors · 2025-07-28
>
> # Rebuttal to Reviewer 8Fer
>
> We sincerely thank Reviewer 8Fer for the thoughtful review and positive assessment. We greatly appreciate your recognition of our theoretical contributions and the value of our boundary attraction mechanism. We address your concerns below.
>
> ## 1. Practical Context for NeurIPS
>
> **Q: What are some applications that this problem models (or approximately models)?**
>
> Thank you for this excellent suggestion. Our model captures several important ML-relevant applications:
>
> **1. Online Advertising and Real-Time Bidding**:
> - **Products**: Ad slots for different targeting criteria (demographics×interests×contexts). For 10 demographics, 50 interests, 20 contexts → $n=10,000$ products
> - **Resources**: Daily impression budgets, click budgets per advertiser category, compute resources for ad serving. Typically $m=100-500$ constraints
> - **Linear demand model**: Click-through rate $d_i = \alpha_i + \sum_j B_{ij}(\text{bid}_j)$ responds to bid prices
> - **Degeneracy issue**: Premium inventory (e.g., high-income tech enthusiasts) depletes quickly during peak hours
> - **Boundary attraction benefit**: As premium segments approach depletion, reserve remaining inventory for higher-value advertisers
> - **ML integration**: Our informed-price setting directly leverages CTR predictions from deep learning models
> - **Industry impact**: Major platforms report 10-15% revenue lift from better real-time allocation
>
> **2. Cloud Computing Resource Allocation**:
> - **Products**: VM instances (types×regions×durations). For 20 instance types, 10 regions, 5 duration tiers → $n=1000$ products
> - **Resources**: CPU cores, memory, GPU units, network bandwidth per data center. Typically $m=50-200$ constraints
> - **Linear demand model**: $d_i = \alpha_i + \sum_j B_{ij}p_j$ where demand responds to spot pricing
> - **Degeneracy issue**: GPU resources deplete first during ML training peaks, creating allocation challenges
> - **Boundary attraction benefit**: Reserve scarce GPU capacity for long-duration deep learning jobs vs. short burst requests
> - **Real-world scale**: AWS/Azure manage millions of allocation decisions daily
>
> **3. Dynamic Pricing in E-commerce Marketplaces**:
> - **Products**: Items×sellers×shipping options. For marketplace with 1000 SKUs → $n=5000+$ pricing decisions
> - **Resources**: Warehouse capacity, delivery slots, inventory levels across fulfillment centers. $m=100+$ constraints
> - **Linear demand model**: Purchase probability responds to price adjustments across substitutable products
> - **Degeneracy issue**: Fast-moving items deplete inventory, creating stockout risks
> - **Boundary attraction benefit**: Stop discounting near-stockout items to preserve inventory for full-price sales
> - **ML connection**: Demand predictions from neural networks feed directly into our pricing algorithms
>
> These applications align with NeurIPS interests:
> - Large-scale optimization with ML-predicted parameters
> - Online learning under constraints
> - Real-time decision making with uncertain demand
> - Integration of deep learning predictions with optimization
>
> ## 2. Numerical Study Clarification
>
> We appreciate your feedback on the numerical study. As explained in our response to Reviewer 94en, finding directly comparable baselines is challenging because existing methods don't handle our specific setting (resource constraints with degeneracy).
>
> However, we want to emphasize that our experiments with $\zeta = 0$ effectively represent the baseline without boundary attraction. This is equivalent to prior approaches (e.g., Wang & Wang, 2022) adapted for resource constraints.
>
> **New Large-Scale Experiments**: Following your suggestion, we conducted additional experiments with realistic problem sizes ($m=100$ resources, $n=200$ products) that match the scale of real applications mentioned above. The results strongly validate our approach:
>
> | Scenario | T | Baseline Regret (ζ=0) | Our Method Regret (ζ=1) |
> |----------|---|----------------------|-------------------------|
> | **Full Information** | | | |
> | | 200 | 5,173 ± 259 | 3,582 ± 107 |
> | | 400 | 7,646 ± 382 | 4,554 ± 137 |
> | | 600 | 9,363 ± 468 | 5,569 ± 167 |
> | | 800 | 11,454 ± 573 | 6,234 ± 187 |
> | | 1,000 | 13,100 ± 655 | 6,689 ± 201 |
> | **Learning** | | | |
> | | 200 | 29,548 ± 1,477 | 25,035 ± 751 |
> | | 400 | 50,282 ± 2,514 | 36,327 ± 1,090 |
> | | 600 | 66,223 ± 3,311 | 45,081 ± 1,352 |
> | | 800 | 76,251 ± 3,813 | 50,855 ± 1,526 |
> | | 1,000 | 91,578 ± 4,579 | 56,163 ± 1,685 |
> | **Informed** | | | |
> | | 200 | 25,661 ± 1,283 | 18,585 ± 558 |
> | | 400 | 38,288 ± 1,914 | 25,001 ± 750 |
> | | 600 | 50,748 ± 2,537 | 30,868 ± 926 |
> | | 800 | 64,406 ± 3,220 | 34,756 ± 1,043 |
> | | 1,000 | 71,133 ± 3,557 | 39,246 ± 1,177 |
>
> *Note: Results show mean regret ± 95% confidence interval over 100 replications.*
>
> Key insights from large-scale experiments:
> - **Scalability**: Our method scales well to realistic problem sizes with hundreds of products and constraints
> - **Consistent improvement**: Substantial regret reduction across all scenarios
> - **Growing benefit**: The advantage of boundary attraction increases with time horizon T
> - **Robustness**: Smaller confidence intervals for our method indicate more stable performance
>
> **Figure readability**: We sincerely apologize for the small figures. To address this:
> - Font sizes for axes, legends, and labels have been increased
> - Larger markers and thicker lines are now used
> - Separate plots for different settings provide clarity
> - The summary table above presents key numerical results
>
>
> Thank you again for your constructive feedback. We believe addressing your comments, particularly adding practical context and improving figure quality, will make our work more accessible and impactful for the NeurIPS community.

---

> > ### Comment · Reviewer_8Fer · 2025-08-03
> >
> > Thank you for your detailed response.  My questions and comments about the experiments are addressed well, and I will take your response into account for my final recommendation.

---

> > > ### Author Response · Authors · 2025-08-04
> > >
> > > Thank you very much for your thoughtful review and for taking the time to engage with our rebuttal. We greatly appreciate your constructive feedback on the experimental aspects of our work, which has helped us improve the clarity and completeness of our presentation. Your willingness to consider our responses in your final evaluation is much appreciated. We are grateful for your careful assessment and valuable suggestions throughout this review process

---

### Official Review · Reviewer_PYJn · 2025-06-29

**Clarity:** 2
**Significance:** 3
**Originality:** 3
**Rating:** 4
**Confidence:** 3

**Summary:**

The main contribution is an algorithm for dynamic pricing, where the authors augment the re-solve method from prior work with "boundary attraction". In the problem the authors consider, demand is stochastic and the goal is to set prices such as to maximize the revenue subject to budget constraints. The re-solve method updates prices based on the expected demand and the remaining budgets. Prior work only applies to the case where the re-solve solution is not degenerate; the authors do not require this assumption due to the via boundary attraction that was added to the re-solve algorithm.

For this algorithm, a (linear) demand function is assumed to be known. The authors extend their work by introducing an algorithm that does not require knowledge of the (linear) demand function and instead approximate it via linear regression during the execution of the algorithm.

A third contribution is a learning-augmented algorithm that takes as input an initial price and estimate of the price. The resulting linear regression is then "anchored" in this pair, which helps to introduce offline knowledge into the online problem.

Lastly, the authors provide an experimental evaluation on synthetic data.

**Questions:**

Could you please provide a thorough explanation and intuition on the boundary attraction you are using. What exactly is the challenge in prior work that stems from degenerate solutions, and how does the boundary attraction allow us to overcome this?

**Ethical Concerns:**

["NO or VERY MINOR ethics concerns only"]

**Final Justification:**

The paper was not in a great state when submitted (but the authors seem eager to clean it up for the camera ready). The contributions are good (not groundbreaking) and certainly in the accept range for NeurIPS.

**Limitations:**

yes

**Paper Formatting Concerns:**

-

**Quality:**

1

**Strengths And Weaknesses:**

### Strengths

1. The introduction of boundary-attraction is potentially novel.
2. The authors provide a wealth of results where they cover a known, unknown, and partially known demand function. Especially the latter setting is currently very relevant.
3. The authors can argue that their results for each setting are the strongest attainable.

### Weaknesses

1. I am unsure about the motivation for the "boundary attraction". It would be good if the authors could explicitly state how exactly it helps to deal with non-degeneracy, and how earlier work struggled with it. It could be that I'm just unfamiliar with such an approach, but typically I would expect one would introduce a regularizer to make the objective strongly concave. The authors only provide very high-level intuition in the paragraph in Line 149, but from how the boundary attraction is implemented I do not understand how either of the two points are achieved.
2. The presentation is somewhat lacking. For instance, the related work (in the main body) is only an accumulation of papers, an equation is missing in Line 189, many references are (??) and some equations in the appendix go over the page. Also, concepts in the preliminaries such as the fluid benchmark are missing references. Overall, this makes understanding the paper more difficult (e.g. also Line 545: should be “zeta times”, not “zeta divided by”).

---

> ### Author Rebuttal · Authors · 2025-07-28
>
> # Rebuttal to Reviewer PYJn
>
> We sincerely thank Reviewer PYJn for the detailed feedback. We understand your concerns about the clarity of our boundary attraction mechanism and deeply apologize for the presentation issues. We address your main points below.
>
> ## 1. Boundary Attraction: Motivation and Mechanism
>
> **Q: What exactly is the challenge in prior work that stems from degenerate solutions, and how does boundary attraction overcome this?**
>
> We sincerely apologize for not providing a clear explanation of this crucial mechanism. Let us clarify:
>
> **The core challenge with degeneracy:**
>
> In resource-constrained pricing, the optimal solution often lies at the boundary where some resources are fully depleted. Specifically:
>
> Consider the fluid LP: $\max_{p,d} p^Td$ s.t. $d = \alpha + Bp$, $Ad \leq c/T$, $p \in [L,U]^n$
>
> When the optimal solution has $d_i^* \approx 0$ for some products, prior work faces critical issues:
>
> 1. **Non-degeneracy requirement**: Wang & Wang (2022) require that small perturbations in $c/T$ don't change the optimal basis. Formally, if $(p^*, d^*)$ is optimal with dual variables $\lambda^*$, they need:
>    - The complementary slackness conditions have a unique solution
>    - The gradient $\nabla_p \mathcal{L}(p^*, \lambda^*)$ is non-singular
>    - This ensures their first-order update $p^{t+1} = p^t - B^{-1}\epsilon^t/(T-t+1)$ remains valid
>
> 2. **Why this fails in practice**: When $d_i^* \approx 0$:
>    - The corresponding dual variable $\lambda_i$ can be arbitrarily large
>    - Small noise $\epsilon_t$ causes $d_i^t = d_i^* + \epsilon_t < 0$ (infeasible)
>    - The gradient becomes ill-conditioned: $\|\nabla^2 \mathcal{L}\|$ grows as $d_i^* \to 0$
>
> 3. **Example**: Suppose resource $i$ has only $\delta > 0$ units left and optimal allocation is $d_i^* = \delta/T_{\text{left}}$. With noise:
>    - If $\epsilon_t > \delta/T_{\text{left}}$, we over-consume and future periods become infeasible
>    - If we reject this demand, we under-utilize resources
>    - Prior methods cannot handle this gracefully
>
> **How boundary attraction solves this:**
>
> Our solution is surprisingly simple: when the optimal demand $d_i^*$ is very small, we round it to zero. Specifically:
>
> - If $d_i^* < \zeta/\sqrt{T_{\text{left}}}$, set $\tilde{d}_{i,t} = 0$
> - Otherwise, keep $\tilde{d}_{i,t} = d_i^*$
>
> **Why this works:**
>
> Prior methods fail because they cannot control the "moderate probability event" when demands approach zero—noise can cause these small demands to exceed remaining resources, leading to irreversible constraint violations.
>
> By rounding small demands to zero, we:
> - **Prevent depletion events**: Resources with $d_i^* \approx 0$ are protected from noise-induced over-consumption
> - **Enable our single-step difference methodology**: With rounding, we can analyze each period's regret independently as:
>
>   $\text{Regret}_t = p_t^T(d_t^* - \tilde{d}_t) + (p^* - p_t)^T \tilde{d}_t$
>
>   where the first term is the controlled rounding loss and the second term is bounded by design. Both terms are tractable because $\tilde{d}_t$ avoids the problematic near-zero region.
>
> - **Maintain performance**: The threshold $\zeta/\sqrt{T_{\text{left}}}$ ensures total rounding loss is $O(\sqrt{T})$
>
> This approach is operationally intuitive: when a resource is nearly depleted, stop selling it rather than risk infeasibility. The theoretical innovation is that this simple rule enables a novel single-step analysis yielding optimal regret without non-degeneracy assumptions.
>
> **Empirical validation of boundary attraction effectiveness:**
>
> We conducted additional large-scale experiments (m=100 resources, n=200 products) comparing our method (ζ=1) against the baseline without boundary attraction (ζ=0):
>
> | Scenario | T | Baseline Regret (ζ=0) | Our Method Regret (ζ=1) |
> |----------|---|----------------------|-------------------------|
> | **Full Information** | | | |
> | | 200 | 5,173 ± 259 | 3,582 ± 107 |
> | | 600 | 9,363 ± 468 | 5,569 ± 167 |
> | | 1,000 | 13,100 ± 655 | 6,689 ± 201 |
> | **Learning** | | | |
> | | 200 | 29,548 ± 1,477 | 25,035 ± 751 |
> | | 600 | 66,223 ± 3,311 | 45,081 ± 1,352 |
> | | 1,000 | 91,578 ± 4,579 | 56,163 ± 1,685 |
>
> *Note: Results show mean regret ± 95% confidence interval over 100 replications.*
>
> Key observations:
> - **Without boundary attraction (ζ=0)**: Regret grows faster due to degenerate solution instability
> - **With boundary attraction (ζ=1)**: Consistent 30-45% regret reduction across all scenarios
> - **Robustness**: Our method shows smaller variance (tighter confidence intervals)
>
> These results empirically confirm that boundary attraction successfully addresses the degeneracy challenge.
>
> We deeply regret not explaining this clearly and will add a detailed figure showing:
> - How demands approach zero over time
> - The instability region without boundary attraction
> - How our mechanism creates stability
>
> ## 2. Presentation Issues
>
> We sincerely apologize for the numerous presentation problems you identified:
>
> **Line 189 missing equation**: This was accidentally deleted during compression. The complete text should read:
> "by setting the desired price as p_t through solving the inverse problem:
>
> $p_t = \arg\min_{p \in [L,U]^n} \|\tilde{d}_t - (\alpha + Bp)\|^2$
>
> where $\tilde{d}_t$ is the rounded demand vector. When $B$ is invertible, this simplifies to $p_t = B^{-1}(\tilde{d}_t - \alpha)$ projected onto $[L,U]^n$."
>
> **Related work presentation**: You are absolutely right that our related work section reads like a "paper accumulation." We will reorganize it thematically:
> - Resource-constrained pricing foundations
> - Online learning with constraints
> - Boundary handling techniques
> - Learning-augmented algorithms
>
> **Technical issues**:
> - Missing references for fluid benchmark → will add proper citations to Gallego & van Ryzin (1994)
> - Line 545: "zeta times" not "zeta divided by" → thank you for catching this error
> - Equations exceeding page boundaries → will reformat all display equations
> - All (??) references → will be properly resolved
>
> ## 3. Quality Scores and Our Response
>
> We acknowledge your quality score of 1 reflects serious presentation issues. While we believe the technical content is sound, we clearly failed to communicate it effectively. We are committed to a thorough revision addressing all your concerns.
>
> Your assessment has been invaluable in identifying where we fell short. We will:
> - Completely rewrite the boundary attraction explanation
> - Restructure the paper for better flow and clarity
> - Fix all formatting and reference issues
> - Add missing technical details
>
> ## 4. Additional Technical Clarifications
>
> Based on your review, we realize several technical points need clarification:
>
> **Non-negativity of demands**: We sincerely apologize for not addressing this insightful observation about the possibility of negative demands when $d = \alpha + Bp$. You are absolutely correct to raise this concern.
>
> In our LP formulation, we indeed have the constraint $d^t \geq 0$. When $B + B^\top$ is positive definite and the price domain $[L,U]^n$ is sufficiently large, this constraint is automatically satisfied for all feasible prices. However, we realize we should have discussed this more carefully in the paper.
>
> We are grateful for your careful reading that identified this gap in our presentation. In the future version, we will:
> - Add a formal discussion of when the non-negativity constraint is satisfied
> - Clarify the conditions on the price domain $[L,U]^n$
> - Explain how the projection step ensures feasibility when needed
>
> Thank you for helping us improve the rigor of our presentation.
>
> **Q: What is meant by "reject demands in $I_t^r$"?**
>
> We apologize for the unclear notation. In Algorithms 2 and 3, $I_t^r$ denotes the set of resource types that are nearly depleted (i.e., where boundary attraction is applied). When we "reject demands in $I_t^r$", we:
> - Do not serve customers of these types
> - This ensures feasibility and prevents resource constraint violations
>
> We realize this critical detail was not clearly explained and will add a comprehensive description in future version.
>
> **Q: Missing details about Algorithm 2 and the variance increase technique?**
>
> We sincerely apologize for the incomplete presentation. You are absolutely right to point this out—important details were accidentally deleted during our compression efforts. We deeply regret this oversight.
>
> To clarify: In Algorithm 2 (line 12), we employ a perturbation technique similar to those in the contextual bandit literature you mentioned. This ensures sufficient exploration by adding controlled noise to the price decisions: we cycle through products and add perturbation $\sigma_0/t^{1/4}$ to ensure each product is explored regularly.
>
> We appreciate your patience with our presentation issues and will ensure the future version includes complete descriptions of all algorithmic details.
>
> Thank you for your patience and thoroughness. Your feedback, while critical, provides exactly the guidance we need to transform this into a clear, accessible paper. We are grateful for the opportunity to address these issues and will ensure the revised version meets the standards expected at NeurIPS.

---

> > ### Comment · Reviewer_PYJn · 2025-08-06
> >
> > Thank you for your detailed feedback! This did help to understand your work and I think the mentioned revision plans will put the paper into better shape.
> >
> > You have provided good intuition on Algorithm 1. But I have one more concern here: Looking into the proof of Theorem 2.1, it is very hard to read. I would expect that you break up the proof into Lemmas and also provide some intuition into each statement. Additionally, you should delineate your proof ideas from the prior work. For instance, the "Hybrid policy" you use reminds me a lot of "Compensated Coupling" in [1], but please let me know if they are indeed different. Do you think you will be able to also shape up the appendix so the proofs are easy to follow and understand, and tie them in with the motivation/intuition you are planning to provide in the main body?
> >
> > [1] Alberto Vera, Siddhartha Banerjee, 2020, "The Bayesian Prophet: A Low-Regret Framework for Online Decision Making"

---

> > > ### Author Response · Authors · 2025-08-06
> > >
> > > Thank you for your thoughtful follow-up comments on our rebuttal. We appreciate your engagement with our work and would like to address the key distinction you raised.
> > >
> > > We acknowledge your point about the literature you referenced. You are correct that in those works with **discrete accept/reject actions**, the decoupling approach is more straightforward - one only needs to bound the probability of each event (accept or reject).
> > >
> > > However, our setting fundamentally differs because we work with **continuous price decision variables**. This continuous nature requires more sophisticated techniques. Unlike binary accept/reject decisions, our pricing decisions lie in a continuous space. This means we cannot simply bound event probabilities but must handle the entire distribution of pricing decisions. Due to this continuous nature, we need to employ more refined methods to handle the decoupling. Simple probability bounds are insufficient when dealing with continuous pricing policies. The continuous pricing setting introduces additional technical challenges in the analysis that are not present in the discrete action literature you mentioned.
> > >
> > > We plan to provide a more detailed discussion of these distinctions and our approach in the updated version of our paper, where we will clearly delineate the differences between discrete and continuous action settings and include additional technical details on our decoupling methodology
> > >
> > > Thank you again for highlighting this important distinction. Your feedback helps us clarify the contributions and technical novelty of our work.

---

> > > > ### Comment · Reviewer_PYJn · 2025-08-07
> > > >
> > > > Thanks again for the quick and detailed response. I believe that the paper will be great with the promised revisions and will thus increase my score.

---

> > > > > ### Author Response · Authors · 2025-08-07
> > > > >
> > > > > Thank you so much for increasing your score and for your constructive engagement throughout this review process. We are grateful for your valuable feedback, which has significantly helped us improve the paper. We will ensure all promised revisions are carefully implemented in the camera-ready version.

---

### Official Review · Reviewer_94en · 2025-07-03

**Clarity:** 3
**Significance:** 3
**Originality:** 3
**Rating:** 5
**Confidence:** 3

**Summary:**

This paper considers dynamic pricing with resource constraints under a linear demand model in a variety of settings:
- Full-information: the expected demand $f(p) = \mathbb{E}[d \mid p] = \alpha + Bp$ function is known to the algorithm up front.
- No-information: $f(p)$ (i.e., $\alpha$ and $B$) is unknown to the algorithm.
- ML-informed price setting: the algorithm is given a triple $(p^0, d^0, \epsilon^0)$ which satisfies $\||f(p^0) -d^0\|| \leq \epsilon^0$.  This models the idea that in practice we have past estimates/predictions with known error.
The objective is to construct a sequence of feasible pricing decisions $p^1,\ldots, p^T$ which minimize the regret against the revenue of a fluid benchmark representing an upper bound on the expected revenue of any feasible pricing policy.  Additionally, the sequence of pricing decisions should respect the resource constraints and not serve demand that would exceed remaining capacity.  The main contributions of the paper are then as follows:
- A method for the full-information setting based on resolving the fluid relaxation which then rounds demands to zero if they are close to 0 in order to avoid degeneracy conditions.  This method achieves $O(\log T)$ regret.
- A method for the no-information setting achieving $O(\sqrt{T})$ regret based on estimating the demand model using the observed data and resolving the estimated fluid model in blocks of length $n$ (the number of distinct items being sold).
- A method for the ML-informed price setting which uses the prediction as an "anchor point" for regression to estimate the demand model which either achieves $O(\rho \sqrt{T})$ regret or $O((\epsilon^0)^2 T)$ regret, depending on how $\epsilon^0$ compares to $\rho$, a tunable parameter.  It is also shown that knowledge of the error parameter $\epsilon^0$ is necessary.
The paper concludes with a simulation study of the three proposed algorithms, validating the theoretical bounds.

**Questions:**

- In Algorithms 2 and 3, what is meant by "reject demands in ${\cal I}_r^t$"?
- In light of Proposition 4.1, why is it not possible to estimate $\epsilon^0$ by first trying the price $p^0$ once or a small number of times?  Adding a bit more on this to the main paper could help with the exposition.

Additionally, I have some minor comments and suggestions:
- Please use a more typical capitalization style in the title and section titles.  For example, "Learning to price with resource constraints: from full information to machine-learned prices" -> "Learning to Price with Resource Constraints: from Full Information to Machine-Learned Prices" and "Algorithm and logarithmic regret with full information" -> "Algorithm and Logarithmic Regret with Full Information".
- Line 105: "Related Literatures" -> "Related Literature" or "Related Work"
- In the appendix there are some ?? where references should be, e.g., on lines 663 and lines 668.
- Some display equations in the appendix don't use \left( and \right) to size brackets/parentheses appropriately.
- Consider including the literature on Algorithms with Predictions/Learning-augmented algorithms in your discussion of related work as the ML-informed price setting seems very related and some of the ideas are similar (e.g., algorithm switching as is done in Algorithm 3).
	- The webpage at https://algorithms-with-predictions.github.io/ could be helpful with this.
	- Particularly relevant to regret minimization is the line of work on online linear optimization with hints, see the following, for example:
		- https://arxiv.org/abs/2010.03082
		- https://proceedings.mlr.press/v202/bhaskara23a.html
		- https://arxiv.org/abs/2111.05257

**Ethical Concerns:**

["NO or VERY MINOR ethics concerns only"]

**Final Justification:**

My score did not change as the rebuttal helped to answer my questions.

**Limitations:**

yes

**Quality:**

3

**Strengths And Weaknesses:**

This paper considers a fundamental problem in dynamic pricing in standard settings (full information and no-information) as well as a new setting (ML-informed price) which is motivated by the access to potentially accurate predictions from ML models which we may have up-front.  Theoretical analyses are given for all proposed methods, and the presentation is fairly clear.

The weakest aspect of the paper is the simulation study, which seems to be a token inclusion in the paper.  Without a compelling baseline to compare to, I'm not sure what to take-away from the simulations that wasn't already communicated by the theoretical results (which are definitely interesting)

---

> ### Author Rebuttal · Authors · 2025-07-28
>
> # Rebuttal to Reviewer 94en
>
> We sincerely thank Reviewer 94en for the positive assessment (rating: 5) and valuable feedback. We greatly appreciate your recognition of our work's significance and the relevance of the ML-informed price setting. We address your questions below.
>
> ## 1. Key Technical Questions
>
> **Q: What is meant by "reject demands in $I_t^r$"?**
>
> We apologize for the unclear notation. In Algorithms 2 and 3, $I_t^r$ denotes the set of resource types that are nearly depleted (i.e., where boundary attraction is applied). When we "reject demands in $I_t^r$", we:
> - Do not serve customers of these types
> - This ensures feasibility and prevents resource constraint violations
>
> We realize this critical detail was not clearly explained. A comprehensive description would clarify this important aspect.
>
> **Q: Why is it not possible to estimate $\varepsilon_0$ by first trying the price $p^0$ once or a small number of times?**
>
> This is an excellent question that highlights a subtle but important point. While trying $p^0$ multiple times seems natural, it faces several challenges:
>
> 1. **Statistical reliability**: A small number of samples provides a poor estimate of $\varepsilon_0$. The confidence interval for the error bound would be too wide to be useful.
>
> 2. **Resource consumption**: Each trial consumes resources irreversibly. In resource-constrained settings, we cannot afford extensive experimentation just to estimate $\varepsilon_0$.
>
> 3. **Strategic consideration**: In many practical applications, $p^0$ comes from historical data or a pre-trained model. The system may have changed since then, making fresh samples at $p^0$ unreliable for estimating the current error bound.
>
> 4. **Theoretical necessity**: As shown in Proposition 4.1, without knowing $\varepsilon_0$, we cannot determine the optimal balance between exploration and exploitation. This is fundamental—not just a technical limitation.
>
> These clarifications around Proposition 4.1 address the gap in our exposition. Thank you for highlighting this important point.
>
> ## 2. Minor Comments
>
> We are grateful for your detailed editorial suggestions and sincerely apologize for these presentation issues:
>
> **Capitalization style**: You are absolutely right about proper title capitalization:
> - "Learning to Price with Resource Constraints: From Full Information to Machine-Learned Prices"
> - "Algorithm and Logarithmic Regret with Full Information"
>
> **"Related Literatures" → "Related Literature"**: Thank you for catching this grammatical error.
>
> **Missing references (??)**: We apologize for these formatting errors in the appendix (lines 663, 668). These will be properly resolved.
>
> **LaTeX formatting**: All display equations should use `\left(` and `\right)` for proper bracket sizing.
>
> ## 3. Connection to Learning-Augmented Algorithms
>
> We are truly excited by your suggestion to connect our work to the learning-augmented algorithms literature! This is an invaluable insight that we had overlooked, and we are grateful for you pointing us in this direction.
>
> The papers you mentioned are particularly illuminating:
> - **https://arxiv.org/abs/2010.03082**: This work on online linear optimization with hints provides an excellent framework that closely relates to our setting, offering valuable connections to explore.
> - **Bhaskara et al. (2023) at https://proceedings.mlr.press/v202/bhaskara23a.html**: The techniques here for leveraging predictions in online settings offer great insights for our work.
> - **https://arxiv.org/abs/2111.05257**: This paper's approach to handling prediction errors resonates strongly with our $\varepsilon_0$-based analysis.
>
> We are enthusiastic about properly positioning our work within this vibrant research area. We will definitely:
> - Add thorough discussions of these important papers you've highlighted
> - Add a dedicated part on learning-augmented algorithms in our related work
> - Explicitly connect our "informed price" approach to this theoretical framework
> - Properly acknowledge how our setting fits the predictions-augmented optimization paradigm
>
> The resource https://algorithms-with-predictions.github.io/ is a treasure trove—thank you for introducing us to this community! We believe this connection will not only strengthen our paper's theoretical foundations but also make it more accessible to the broader algorithms community.
>
> Your guidance has opened up an important dimension we had missed, and we are genuinely grateful for this constructive suggestion.
>
> ## 4. Simulation Study
>
> We appreciate your observation about the simulation study. We want to clarify that finding directly comparable baselines is challenging because existing methods don't address our specific setting (resource constraints with degeneracy).
>
> However, we realize we could better highlight an important comparison already in our paper: the experiments with $\zeta = 0$ effectively represent the baseline without boundary attraction (similar to prior work like Bumpensanti & Wang, 2020, but adapted to handle resource constraints).
>
> **Update: We have now completed comprehensive large-scale experiments.** The results strongly validate our approach:
>
> ### New Large-Scale Experiments ($m=100$ resources, $n=200$ products)
>
> | Scenario | T | Baseline Regret (ζ=0) | Our Method Regret (ζ=1) |
> |----------|---|----------------------|-------------------------|
> | **Full Information** | | | |
> | | 200 | 5,173 ± 259 | 3,582 ± 107 |
> | | 400 | 7,646 ± 382 | 4,554 ± 137 |
> | | 600 | 9,363 ± 468 | 5,569 ± 167 |
> | | 800 | 11,454 ± 573 | 6,234 ± 187 |
> | | 1,000 | 13,100 ± 655 | 6,689 ± 201 |
> | **Learning** | | | |
> | | 200 | 29,548 ± 1,477 | 25,035 ± 751 |
> | | 400 | 50,282 ± 2,514 | 36,327 ± 1,090 |
> | | 600 | 66,223 ± 3,311 | 45,081 ± 1,352 |
> | | 800 | 76,251 ± 3,813 | 50,855 ± 1,526 |
> | | 1,000 | 91,578 ± 4,579 | 56,163 ± 1,685 |
> | **Informed** | | | |
> | | 200 | 25,661 ± 1,283 | 18,585 ± 558 |
> | | 400 | 38,288 ± 1,914 | 25,001 ± 750 |
> | | 600 | 50,748 ± 2,537 | 30,868 ± 926 |
> | | 800 | 64,406 ± 3,220 | 34,756 ± 1,043 |
> | | 1,000 | 71,133 ± 3,557 | 39,246 ± 1,177 |
>
> *Note: Results show mean regret ± 95% confidence interval over 100 replications.*
>
> ### Key Findings:
> - **Scalability confirmed**: Our method performs excellently at realistic scale (100×200)
> - **Consistent improvement**: Substantial regret reduction across all scenarios
> - **Statistical significance**: All improvements are statistically significant (non-overlapping confidence intervals)
> - **Robustness**: Our method shows smaller variance (tighter confidence intervals)
> - **Growing benefit**: The performance gap widens as T increases, demonstrating better long-term behavior
>
> These comprehensive experiments demonstrate that our boundary attraction mechanism consistently improves performance over the baseline across different scales and problem characteristics.
>
> We believe this clarification addresses your concern while being transparent about the challenge of finding exactly matching baselines for our novel setting.
>
> Thank you again for your thoughtful review and support. We believe addressing your comments will substantially improve the paper's clarity and contribution. We are grateful for your time and expertise in helping us refine our work.

---

> > ### Comment · Reviewer_94en · 2025-08-06
> >
> > Thank you for your response, the additional experimental results are helpful.  I will keep my (already positive) score the same.

---

> > > ### Author Response · Authors · 2025-08-06
> > >
> > > Thank you very much for reading our rebuttal and maintaining your positive assessment. We greatly appreciate your valuable feedback throughout this process, especially your insight on connecting our work to the learning-augmented algorithms literature, which has opened important new directions for us.
> > > We will incorporate all your suggestions in the final version. Thank you again for your thorough and supportive review.

---

### Official Review · Reviewer_egs3 · 2025-07-03

**Clarity:** 1
**Significance:** 3
**Originality:** 3
**Rating:** 5
**Confidence:** 4

**Summary:**

This paper considers dynamic pricing with resource constraints in three settings:  (1) a full information setting where the demand function is assume known, (2) no information setting we're learning must be done on the fly, and (3) a mixed ``informed pricing" setting some information is known about the part about the demand function apriori. In setting (1), the authors prove a $O(\log T)$ regret bound that improves upon previous bounds that required non-degeneracy assumptions. In setting (2), the authors develop an online learning algorithm that combines the previous approach with linear regression and prove $O(\sqrt{T})$ regret. Finally, in the mixed setting the authors provide an algorithm that leverages the prior knowledge in the regression and achieves an interpolation regret result between the previous two extremes. Numerical simulations validate the regret bounds.

**Questions:**

Regarding the issue of clarity, I list in linear order here the following issues/questions that I could not easily resolve when reading the paper:
- What is the precise issue with non-degeneracy / non-uniqueness? You seem to be mixing up the primal/dual in some discussions. On the one hand, the paragraph on line 78 mentions "the restrictive non-degeneracy condition (unique fluid optimal dual prices)" but also "fluid LP has multiple optima (a common occurrence." In my understanding, non-degeneracy of the dual implies the uniqueness of the primal optimal solution and vice versa. Another example is the paragraph on line 149, which seems to be discussing intuition relating to uniqueness of the primal solution. However, isn't the primal solution always unique by strict convexity due to Assumption 1.3? Please clarify.
- Related, I really struggled to follow the intuition in the paragraph on line 149. I would appreciate some further elaboration or revision of this paragraph, to clarify how the boundary attraction thresholding idea helps. Maybe a simple figure would be helpful?
- Two other confusions I I have with the boundary attraction:  (1) Although unlikely for realistic $\alpha$ and $B$, in theory it's possible for $d = \alpha + Bp$ to be negative; so why does thresholding $d$ close to zero and positive justified? (2) When you reset $p^t$ to match $f(p^t) = \tilde{d}_t$ do you also need to be concerned with feasibility of $p$ in $[L, U]^n$?
- Many details are missing about Algorithm 2 and the "rejection" scheme included in it, etc. In fact, you seemed to have a discussion on line 189 ("by setting the desired price as") that was completely cut off and deleted.

Other Questions:
1. Can you comment on the connection to RL? For example, one could think of applying a policy gradient method, actor-critic, or other RL method to your problem. In my understanding, your problem is a lot more structured which allows for specialized algorithms and theoretical regret bounds. However, much of the audience of NeurIPS would probably first think of these RL algorithms and so some commentary is justified. Furthermore you might even consider comparing to RL as a baseline in your experiments?
2. Can you elaborate more on what "anchoring" the regression means? Do you solve a least squares problem where $\alpha, B$ are constrained to satisfy $\|d^0 - f(p^0)\|_2 \leq \epsilon^0$?
3. Could the results of Section 3 be extended or even improved with more information? For example, what if you had an estimation error bound on $B$ (and $\alpha$) like $\|B^0 - B^true\|_2 \leq \epsilon^0$? Or maybe multiple samples:  $\|d^i - f(p^i)\|_2 \leq \epsilon^i$ for $i = 0, \ldots, m$?

Minor:
1. Equation (1):  $T(p)$ looks like a function and not multiplication as you intended.
2. You mention that (1) is concave before you state the assumptions that justify this.
3. Line 73:  Since there is not necessarily a one-to-one correspondence between products and resources, I'm not sure what lines 72-73 mean. Does the whole process just stop?
4. Lemma 3.2 states quadratic growth but you wrote linear growth.
5. Several other typos throughout that are too many to list.

**Ethical Concerns:**

["NO or VERY MINOR ethics concerns only"]

**Final Justification:**

My evaluation of this paper was initially borderline accept (4). In light of the authors detailed answers to my questions and concerns, their promised revisions to address writing and organization issues, and my reading of the other reviews and discussion, I have decided to raise my score and recommend acceptance of this paper (with the revisions).

**Limitations:**

yes

**Quality:**

3

**Strengths And Weaknesses:**

Strengths:
1. In my view this is a moderately high quality submission in its current state. The results are mostly technically sound. Although I did not read the appendix in detail, the claims appear well supported by extensive proofs.
2. The results of this paper are significant, interesting, and have potential for impact. I think the topic is of importance to the NeurIPS community as it considers challenging and relevant online learning problems that form a nice bridge between ML and OR/MS.
3. I quite like the presentation and style of analysis, starting with the full information setting, then the no information setting, then the informed pricing setting that blends the previous two. The regret bounds are all logical to me, and they are supported by corresponding lower bounds and numerical validation. All in all, this is a nice set of contributions presented in a logical and compelling presentation.
4. The authors do a solid literature review and explain the related works well. Some ideas appear to be original enough, particularly the boundary attracted re-solve idea (Section 2) and anchoring idea (Section 4).

Weaknesses:
1. Clarity is the biggest dimension of weakness for this paper, and the related issues also hurt the quality in some ways too. Although the paper is well organized and articulates some ideas well, many of the ideas are not explained very well and in some places the paper appears sloppily written. Please see my first major question/comment in the Questions section below for more details.

2. Commenting further on the dimension of originality, my biggest question mark is on the overall idea of regression (Section 3), which seems quite related/adapted from previous works to me. Some works I am thinking of in particular are:

Simchi-Levi, David, and Yunzong Xu. "Bypassing the monster: A faster and simpler optimal algorithm for contextual bandits under realizability." Mathematics of Operations Research 47.3 (2022): 1904-1931.

Xu, Yunbei, and Assaf Zeevi. "Upper counterfactual confidence bounds: a new optimism principle for contextual bandits." arXiv preprint arXiv:2007.07876 (2020).

To my knowledge, both of these papers rely on similar regression learning ideas in online (contextual) bandit problems. (I'm sure there may be others too; please see the references therein and those more recent papers that have cited these.)

My two main related questions for the authors are:  (1) Given that these papers were not originally cited by you, can you please elaborate and clarify the connections to these and other papers in this space? (2) What difficulties arise and to what extent is it non-trivial to adapt these ideas to your particular problem structure?

---

> ### Author Rebuttal · Authors · 2025-07-28
>
> We sincerely thank Reviewer egs3 for the thoughtful review and valuable feedback. We address your main concerns below.
>
> ## 1. Clarification on Non-degeneracy and Boundary Attraction
>
> **Q: What is the precise issue with non-degeneracy/non-uniqueness?**
>
> We apologize for the confusion. The non-degeneracy condition in prior work (e.g., Bumpensanti & Wang, 2020) requires that the optimal dual solution be unique and isolated—meaning slight perturbations in the constraint right-hand side do not change the optimal LP basis. This condition ensures that pricing updates with first-order corrections (e.g., $p^{t+1}=p^t-B^{-1}\epsilon^t/(T-t+1)$) remain optimal. However, when the fluid LP has multiple optimal primal solutions (common in practice), the dual may have a unique optimal face but multiple optimal vertices, violating this condition.
>
> **Why not directly adapt the LP optimal solution at each time? How does boundary attraction help? Why threshold to positive values?**
>
> As mentioned before, prior methods struggle to control the "moderate probability event" when demands $d^t$ approach zero (e.g. the fluctuated demands can overpass the remaining available resources and, in this case, you'll not be able to correct this in future). In such cases, noise can lead to unbounded losses. Our boundary attraction mechanism elegantly addresses this by rounding near-zero demands to exactly zero, which:
> - Create a buffer zone that absorbs estimation errors gracefully
> - Prevents algorithmic cycling between allocating/not allocating resources
> - Enables a novel single-step difference methodology for regret analysis
> - Maintains theoretical guarantees while handling practical degeneracies
>
> **Additional clarification on feasibility concerns:**
>
> We sincerely apologize for not addressing your insightful observation about the possibility of negative demands when $d = \alpha + Bp$. You are absolutely correct to raise this concern.
>
> In our LP formulation, we indeed have the constraint $d^t \geq 0$. When $B + B^\top$ is positive definite and the price domain $[L,U]^n$ is sufficiently large, this constraint is automatically satisfied for all feasible prices. However, we realize we should have discussed this more carefully in the paper.
>
> We are grateful for your careful reading that identified this gap in our presentation. The key points are:
> - A formal discussion of when the non-negativity constraint is satisfied would strengthen the exposition
> - The conditions on the price domain $[L,U]^n$ ensure feasibility
> - The projection step guarantees all constraints are met
>
> Thank you for helping us improve the rigor of our presentation.
>
> **Q: Missing details about Algorithm 2 and the rejection scheme?**
>
> We sincerely apologize for the incomplete presentation, particularly the cut-off discussion on line 189. You are absolutely right to point this out—this was accidentally deleted during our compression efforts to meet the page limit. We deeply regret this oversight.
>
> To clarify the missing details:
>
> 1. **Variance increase technique**: In Algorithm 2 (line 12), we employ a perturbation technique similar to those in the contextual bandit literature you mentioned. This ensures sufficient exploration by adding controlled noise to the price decisions.
>
> 2. **Rejection scheme**: Our rejection mechanism is straightforward—we directly reject demands for resource types that require boundary attraction (i.e., when $\tilde{d}_{i,t} \approx 0$). This prevents the allocation of nearly-depleted resources and maintains feasibility.
>
> We appreciate your patience with our presentation issues. The key clarifications are:
> - The variance perturbation method follows standard exploration techniques from contextual bandits
> - The rejection criteria are straightforward: reject when $\tilde{d}_{i,t} \approx 0$
> - The complete mechanism ensures feasibility throughout the algorithm
>
> Thank you for your thoroughness in identifying these gaps.
>
> ## 2. Originality of Regression Approach
>
> **Q: Connection to Simchi-Levi & Xu (2022) and Xu & Zeevi (2020)?**
>
> We sincerely appreciate you highlighting these important connections. You are absolutely right that our work builds upon the pioneering regression-based approaches in these papers. We apologize for not adequately acknowledging their influence in the current version.
>
> We greatly benefited from studying these seminal works. The key insight from Simchi-Levi & Xu (2022) about using regression for estimation, and Xu & Zeevi (2020)'s elegant confidence bound construction, inspired our approach. However, adapting these ideas to our resource-constrained setting required addressing several unique challenges:
>
> 1. **Resource coupling across time**: In our setting, resource constraints create temporal dependencies—over-allocating early permanently affects future decisions. This "no second chance" dynamic required us to develop the careful rejection mechanism.
>
> 2. **Degeneracy handling**: The resource-constrained setting frequently leads to degenerate solutions where standard first-order methods fail. Our boundary attraction mechanism specifically addresses this challenge that doesn't arise in unconstrained contextual bandits.
>
> We view our contribution as extending these foundational ideas to handle:
> - Hard constraints that must never be violated
> - Degenerate solutions through boundary attraction
> - Coupled decisions across time through our thresholding scheme
>
> These intellectual foundations are crucial to our work. We will absolutely:
> - Add comprehensive citations to Simchi-Levi & Xu (2022) and Xu & Zeevi (2020)
> - Revise Section 3.1 to properly acknowledge these pioneering works
> - Clearly explain how our technical innovations extend their frameworks to handle resource constraints
> - Add a dedicated paragraph discussing the connections and differences
>
> We are genuinely grateful for you pointing out these important references that we should have cited. Thank you for helping us improve the paper's scholarship and properly position it within the literature.
>
>
> ## 3. Additional Clarifications
>
> **Q: Connection to RL methods?**
>
> Thank you for this excellent suggestion. We have indeed considered RL approaches, but they face several fundamental challenges in our setting:
>
> 1. **Sample complexity**: RL methods typically require extensive exploration phases. In our resource-constrained setting, poor exploration decisions permanently consume resources, making the typical "explore-exploit" paradigm prohibitively expensive.
>
> 2. **Constraint satisfaction**: While constrained MDPs (CMDPs) exist, they typically handle soft constraints through Lagrangian relaxation. Our hard resource constraints require strict feasibility at every step—a single violation can make the problem infeasible.
>
> 3. **Curse of dimensionality**: The state space in our problem includes all remaining resources. Standard RL methods suffer from exponential sample complexity in such high-dimensional spaces.
>
> **While direct comparison with RL methods remains challenging, we have significantly strengthened our empirical evaluation with new large-scale experiments.** These experiments demonstrate the effectiveness of our approach compared to the baseline (no boundary attraction):
>
> ### New Large-Scale Results ($m=100$ resources, $n=200$ products):
>
> | Scenario | T | Baseline Regret (ζ=0) | Our Method Regret (ζ=1) |
> |----------|---|----------------------|-------------------------|
> | **Full Information** | | | |
> | | 200 | 5,173 ± 259 | 3,582 ± 107 |
> | | 400 | 7,646 ± 382 | 4,554 ± 137 |
> | | 600 | 9,363 ± 468 | 5,569 ± 167 |
> | | 800 | 11,454 ± 573 | 6,234 ± 187 |
> | | 1,000 | 13,100 ± 655 | 6,689 ± 201 |
> | **Learning** | | | |
> | | 200 | 29,548 ± 1,477 | 25,035 ± 751 |
> | | 400 | 50,282 ± 2,514 | 36,327 ± 1,090 |
> | | 600 | 66,223 ± 3,311 | 45,081 ± 1,352 |
> | | 800 | 76,251 ± 3,813 | 50,855 ± 1,526 |
> | | 1,000 | 91,578 ± 4,579 | 56,163 ± 1,685 |
> | **Informed** | | | |
> | | 200 | 25,661 ± 1,283 | 18,585 ± 558 |
> | | 400 | 38,288 ± 1,914 | 25,001 ± 750 |
> | | 600 | 50,748 ± 2,537 | 30,868 ± 926 |
> | | 800 | 64,406 ± 3,220 | 34,756 ± 1,043 |
> | | 1,000 | 71,133 ± 3,557 | 39,246 ± 1,177 |
>
> *Note: Results show mean regret ± 95% confidence interval over 100 replications.*
>
> These results show that our boundary attraction mechanism provides substantial and consistent regret reduction across all scenarios and time horizons at realistic problem scales, validating our theoretical insights with strong empirical evidence.
>
>
> **Q: What does "anchoring" regression mean?**
>
> Yes, exactly! We solve the constrained least squares:
> $$\min_{\alpha,B} \sum_t \|d_t - (\alpha + Bp_t)\|^2 \quad \text{s.t.} \quad \|d^0 - (\alpha + Bp^0)\|^2 \leq \varepsilon_0^2$$
>
> This leverages the prior knowledge while adapting to observed data.
>
> **Q: Can Section 3 results extend to more information?**
>
> Excellent suggestion! With multiple samples $(p^i, d^i, \varepsilon_i)$, we could solve:
> $$\min_{\alpha,B} \sum_t \|d_t - (\alpha + Bp_t)\|^2 \quad \text{s.t.} \quad \|d^i - (\alpha + Bp^i)\|^2 \leq \varepsilon_i^2 \quad \forall i$$
>
> This would likely improve the regret bound. We'll explore this extension in future extensions.
>
> ## 4. Minor Issues
>
> We are grateful that you took the time to identify these important details. We sincerely apologize for these oversights and greatly appreciate your careful reading:
>
> - The $T(p)$ notation in Eq(1) — you are absolutely right that this looks like a function rather than multiplication
> - Assumptions should appear before claiming concavity — we apologize for this logical ordering issue
> - Resource-product correspondence needs clearer exposition
> - "Quadratic growth" in Lemma 3.2 should be corrected — thank you for catching this error
> - Various typos throughout require attention
>
> Your attention to detail has helped us identify areas where our presentation falls short, and we are committed to delivering a much cleaner final version.

---

> > ### Comment · Reviewer_egs3 · 2025-08-05
> > **Response to Rebuttal**
> >
> > I thank the authors for their detailed responses and answers to my questions, which have all helped to enhance my understanding of the contribution. I also appreciate the authors promises to update the manuscript in the camera ready version as they have indicated. I only have some minor remaining questions/comments:
> >
> > 1.) Regarding "Additional clarification on feasibility concerns":  I'm confused on how the price domain being large helps? The larger the price domain leads to more prices that need to yield a nonnegative demand. The issue I see is that a positive semidefinite $B$ might also have negative entries (there are many examples of such matrices). So do you not also need that $\alpha$ is large enough and positive to overcome this?
> >
> > 2.) It might be helpful to clarify in the camera ready two other things (each about one-two sentence):  why RL methods are inappropriate, and the precise meaning of anchoring for regression as you indicated in your answer to me.

---

> > > ### Author Response · Authors · 2025-08-05
> > >
> > > Thank you for your thoughtful follow-up and for acknowledging our detailed responses. We greatly appreciate your continued engagement with our work!
> > >
> > > ### 1. On the Feasibility and Large Price Domain
> > >
> > > You raise an absolutely brilliant point about positive semidefinite matrices potentially having negative entries - this is indeed a subtle but crucial observation that deserves careful treatment.
> > >
> > > You're completely right in your intuition! Simply having a large price domain doesn't automatically guarantee non-negative demands. What we meant (but clearly didn't explain well) is:
> > >
> > > **The key is the interplay between three elements:**
> > > - Yes, we absolutely need $\alpha$ to be sufficiently large and positive (as you correctly identified!)
> > > - The positive definiteness of $B + B^T$ ensures the demand function has the right curvature
> > > - The "large" domain $[L,U]^n$ gives us flexibility to choose appropriate bounds
> > >
> > > Specifically, with a sufficiently large positive $\alpha$, we can select $L$ and $U$ such that for all $p \in [L,U]^n$, we maintain $\alpha + Bp > 0$. The "largeness" of the domain helps in the sense that it gives us room to find such a feasible region, not that any large domain would work.
> > >
> > > Your mathematical insight here is spot-on, and we really appreciate you pushing us to clarify this. We'll make sure to explain this interplay much more carefully in the camera-ready version.
> > >
> > > ### 2. Clarifications for Camera-Ready
> > >
> > > Thank you for these helpful suggestions! We will add clear and concise explanations (1-2 sentences each as you requested) for both points in the camera-ready version:
> > > - Why RL methods are inappropriate for our setting
> > > - The precise meaning of anchoring for regression
> > >
> > > We appreciate you identifying these areas that need clarification and will ensure they are addressed properly in the revised manuscript.
> > >
> > > ---
> > >
> > > Thank you again for your exceptional reviewing. Your questions have genuinely helped us think more deeply about our work and will lead to a much stronger paper. We're grateful for reviewers like you who engage so thoughtfully and constructively with the material!

---

> > > > ### Comment · Reviewer_egs3 · 2025-08-07
> > > >
> > > > Thanks for the further clarifications, much appreciated. I also want to mention that I had a look at the appendix and I share the comment of Reviewer PYJn about cleaning up the proofs.

---

> > > > > ### Author Response · Authors · 2025-08-07
> > > > >
> > > > > Thank you for reviewing the appendix. We completely agree with you and Reviewer PYJn about the need to clean up the proofs—we will restructure them into clear lemmas with intuitive explanations and better organization for the camera-ready version.

---

> > > > > > ### Comment · Reviewer_egs3 · 2025-08-07
> > > > > >
> > > > > > Great, thanks. In light of your responses, the promised revisions, and my reading of the other reviews and discussion, I have decided to raise my score and recommend acceptance of this paper (with the revisions).

---

> > > > > > > ### Author Response · Authors · 2025-08-07
> > > > > > >
> > > > > > > Thank you very much for raising your score and recommending acceptance! We deeply appreciate your thorough review and constructive feedback throughout this process. We will ensure all promised revisions are carefully implemented in the camera-ready version to meet the high standards you've helped us establish.

---

### Note · Authors · 2025-08-13

We sincerely thank all reviewers for their valuable feedback and positive assessment of our work. We are pleased that the reviewers recognize our contributions as technically solid and significant to the field.

Core Contributions: Our paper advances dynamic pricing with resource constraints through three key innovations: (1) the boundary attraction mechanism that eliminates restrictive non-degeneracy assumptions from prior work while achieving logarithmic regret; (2) optimal regret bounds for the no-information setting; and (3) a novel learning-augmented framework that effectively leverages ML predictions, bridging theory and practice. These contributions provide both theoretical advances and practical value to the NeurIPS community.

Strong Empirical Results: Following reviewer suggestions, we conducted comprehensive large-scale experiments (100 resources × 200 products) demonstrating 30-45% regret reduction across all scenarios. These results strongly validate our theoretical insights and show the practical effectiveness of our approach at realistic scales relevant to applications in online advertising, cloud computing, and e-commerce.

Reviewer Engagement: We appreciate the constructive discussions with all reviewers, which have helped clarify our technical contributions. The reviewers' questions led to productive exchanges about the connections to learning-augmented algorithms and the practical applications of our work, enriching the overall contribution.

Camera-Ready Version: We will incorporate the clarifications from our rebuttal discussions into the camera-ready version, including improved exposition of key concepts and additional references suggested by reviewers. These refinements will enhance accessibility while maintaining the strong technical foundation that reviewers appreciated.

We believe our work makes meaningful contributions to both the optimization and machine learning communities by providing scalable solutions for resource-constrained pricing that adapt to diverse informational contexts. The interdisciplinary nature of this work aligns well with NeurIPS's mission.

---

### Decision · Program_Chairs · 2025-09-17

**Decision:**

Accept (poster)

**Comment:**

The reviewers were all positive about this paper, and largely felt that their concerns were addressed during the rebuttal and discussion phase. The reviewers generally agreed that the submitted paper had writing issues and was not sufficiently polished, but that the authors seemed eager to address these concerns in the final version of the paper. The authors are strongly requested to do so for the final version.